# Token-Based Detection of Spurious Correlations in Vision Transformers

**Solha Kang**  *solha.kang@ghent.ac.kr*
*Ghent University*
*Ghent University Global Campus*

**Esla Timothy Anzaku**  *eslatimothy.anzaku@ghent.ac.kr*
*Ghent University*
*Ghent University Global Campus*

**Wesley De Neve**  *wesley.deneve@ghent.ac.kr*
*Ghent University*
*Ghent University Global Campus*

**Arnout Van Messem**  *arnout.vanmessem@uliege.be*
*Université de Liège*

**Joris Vankerschaver**  *joris.vankerschaver@ghent.ac.kr*
*Ghent University*
*Ghent University Global Campus*

**Francois Rameau**  *francois.rameau@sunykorea.ac.kr*
*State University of New York Korea*

**Utku Ozbulak**  *utku.ozbulak@ghent.ac.kr*
*Ghent University*
*Ghent University Global Campus*
*George Mason University Korea*

**Reviewed on OpenReview:** *https://openreview.net/forum?id=GlPXPhwOzI*

## Abstract

Due to their powerful feature association capabilities, neural network-based computer vision models have the ability to detect and exploit unintended patterns within the data, potentially leading to correct predictions based on incorrect or unintended but statistically relevant signals. These clues may vary from simple color aberrations to small pieces of text within the image. In situations where these unintended signals align with the predictive task, models can mistakenly link these features with the task and rely on them for making predictions. This phenomenon is referred to as spurious correlations, where patterns appear to be associated with the task but are actually coincidental. As a result, detection and mitigation of spurious correlations have become crucial tasks for building trustworthy, reliable, and generalizable machine learning models. In this work, we present a token-based diagnostic pipeline that applies leave-one-out token removal to detect spurious correlations in vision transformers. The proposed approach quantifies a model's reliance on non-core visual cues through complementary measures that capture both aggregate and localized spurious effects at the token level. Using both supervised and self-supervised trained models, we present large-scale experiments on the ImageNet dataset demonstrating the ability of the proposed method to identify spurious correlations. We also find that, even if the same architecture is used, the training methodology has a substantial impact on the model's reliance on spurious correlations. Furthermore, we show that for certain ImageNet classes, many images exhibit

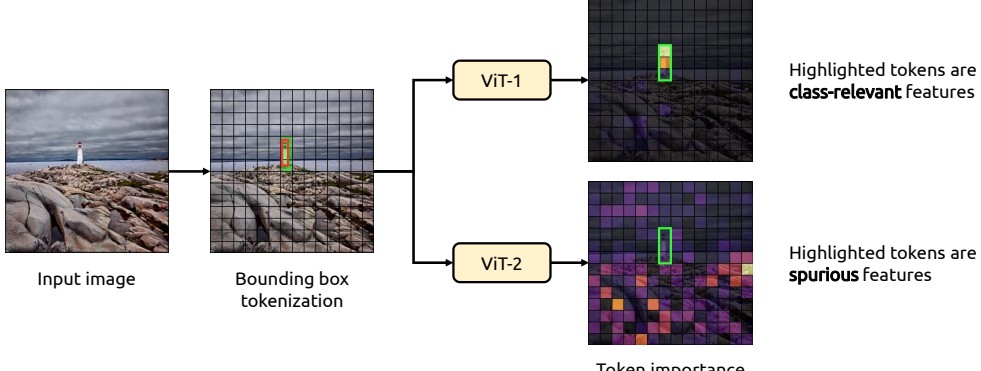

Figure 1: Visual illustration of the approach used in this work to detect spurious correlations. Given an input image, we identify important tokens using a token discarding method and analyze these tokens based on the location of the object of interest. In the first case (ViT-1), the important features lie within the object's bounding box, while in the second case (ViT-2), the important features highlight other elements, indicating the presence of spurious correlations.

strong reliance on non-core visual cues across multiple models, and we discuss common sources of such signals (e.g., watermarks and background artifacts). Lastly, we present a case study investigating spurious signals in invasive breast mass classification, grounding our work in a real-world scenario.

# 1 Introduction

Although the computer vision (CV) community has benefited greatly from the improved performance of neural networks (NNs) on many complex vision tasks (He et al., 2016; Szegedy et al., 2016), recent studies have shown that unintended shortcut learning and spurious correlations are more common than previously thought (Geirhos et al., 2020; Oakden-Rayner et al., 2020). In the context of CV, shortcut learning can be described by what are called core and non-core features (Singla & Feizi, 2021). Here, core features are associated with the desired, semantically meaningful attributes of the image under consideration, whereas non-core features exhibit some degree of correlation with the desired task, while being peripheral or incidental to the main task (Kirichenko et al., 2022). As a result, the prevalence of non-core features may lead to models relying on superficial cues rather than truly understanding the underlying concepts. Analogous concerns have been raised in NLP with the advent of large language models (LLMs): transformer-based models like BERT and GPT-3 have demonstrated extraordinary capabilities, but they too can latch onto dataset artifacts or spurious cues, which can propagate flaws across many downstream applications. Naturally, this phenomenon poses significant challenges to the reliability and generalization capability of NNs across vision and language tasks, highlighting the pressing need for more robust and interpretable learning algorithms in the field (Geirhos et al., 2020).

One of the primary reasons for the existence of non-core features is the tremendous feature association capability of NNs, which can discover and learn signals that are not intended (Geirhos et al., 2020). The introduction of transformer-based models such as the vision transformer (ViT) further exacerbated the aforementioned issue due to the lack of inductive bias and the attention mechanism, which can discover and associate complex signals from even the most distant regions of an image (Vaswani et al., 2017).

Most of the work related to spurious correlations can be categorized into two main areas: (1) robust training methods designed to prevent spurious correlations during the training phase (Gong et al., 2019; Zhou et al., 2022), and (2) post-training detection of spurious correlations on a per-image basis (Lapuschkin et al., 2019; McCoy et al., 2019). The majority of research in the second category relies on interpretability methods, such as GradCAM (Selvaraju et al., 2017), as proxies for identifying spurious correlations. However, these

interpretability methods are known to occasionally highlight misleading or irrelevant regions of the image, raising concerns about their reliability (Adebayo et al., 2018; Kindermans et al., 2019).

To address these issues, we propose a diagnostic pipeline for detecting the presence or absence of spurious correlations in a given image using a trained model. Our method relies on intrinsic properties of vision transformers and leverages the token discarding mechanism to identify spurious correlations through influential tokens that significantly impact the model's predictions (see Figure 2). This token-based strategy relies on the fact that ViTs are capable of processing inputs with variable sequence lengths and builds on recent findings that many patch tokens in ViTs can be removed with minimal effect on model accuracy (Rao et al., 2021; Fayyaz et al., 2022; Bolya et al., 2022). Notably, eliminating such tokens not only speeds up inference but can also make model decisions more interpretable by revealing which regions the model deems important (Rao et al., 2021). This design choice is especially motivated by prior work by Jain et al. (2022), which shows that token removal can better represent feature absence in ViTs than value-replacement baselines, thereby mitigating missingness bias. Building on this insight, our approach repurposes token discarding as a diagnostic tool rather than an efficiency measure: by identifying which tokens a ViT model relies on most for a given image, we can tell whether the model's prediction is based on core features or on potentially spurious cues.

This strategy allows us to assess the trustworthiness of ViTs in terms of their ability to utilize core features and is widely applicable to any transformer-based CV model. Subsequently, we compare the positions of these influential tokens with the provided bounding box information to quantify the extent of spurious correlation using two novel metrics: the Average Token Spuriosity Index (A-TSI) and the Maximum Token Spuriosity Index (M-TSI). A visual overview of the proposed method is provided in Figure 1. Through large-scale experiments on both supervised and self-supervised ViTs on the widely-used ImageNet dataset (Russakovsky et al., 2015), we discover that model training significantly impacts the reliance of models on core and non-core features. Using the proposed approach, we also identify ImageNet classes that contain many images with strong reliance on non-core features across different models, highlighting recurring dataset-level artifacts that models may exploit.

## 2 Related work

Spurious correlations have been investigated in different contexts in the literature, such as invariant learning (Arjovsky et al., 2019), domain generalization (Wang et al., 2022), group robustness (Liu et al., 2021), shortcut learning (Geirhos et al., 2020; Du et al., 2023) and simplicity bias (Tiwari & Shenoy, 2023). Traditionally, most of the research focus has been on training methods which attempt to avoid spurious correlations from being learned, while our method focuses on the detection of spurious correlations after the training is complete, which is a comparatively unexplored area of research.

**Training-based methods to avoid spurious correlations.** Invariant learning methods such as Invariant Risk Minimization (Arjovsky et al., 2019) aim to learn features that remain predictive across multiple training environments. This is typically done by constraining the model to rely on consistent, environment-invariant signals rather than spurious correlations present in individual data distributions. Group robust training focuses on improving model performance across different subgroups within a dataset, particularly those that are underrepresented or prone to high error rates. Optimization strategies include minimizing the loss of the worst-case subgroup loss (Sagawa et al., 2019) and iteratively identifying and correcting errors in hard-to-learn subgroups (Liu et al., 2021), reducing the reliance on shortcut features that perform well only for the majority group. Domain generalization techniques aim to improve model robustness by ensuring that learned features remain representative when shifting from the training dataset to unseen data (Wang et al., 2022). Some approaches achieve this by adversarial training (Ganin et al., 2016; Tzeng et al., 2017), encouraging the model to ignore domain-specific variations, while others promote feature alignment across different datasets (Jin et al., 2020; Lu et al., 2022).

Unlike the aforementioned training-time intervention methods, which actively modify the learning process of models to reduce reliance on spurious correlations, we tackle the problem of post-training identification of spurious correlations.

**Attribution methods to detect spurious correlations.** Another line of research, similar to ours, identifies spurious correlations by investigating the attributions of models to determine whether they are focusing on semantically relevant regions (Ghosal & Li, 2024). Research efforts that follow this approach often analyze prediction changes when specific features are removed using polygon masks or occlusion (Plumb et al., 2021; Zeiler & Fergus, 2014). Region selection is typically guided by interpretability techniques such as GradCAM (Selvaraju et al., 2017), LIME (Ribeiro et al., 2016), and Integrated Gradients (Sundararajan et al., 2017). However, the reliability of these interpretability techniques is often under debate, since their effectiveness depends on assumptions about the underlying model behavior and feature importance (Adebayo et al., 2018; Kindermans et al., 2019). For instance, studies have shown that some attribution methods can produce visually plausible explanations even when applied to randomly initialized networks, raising concerns about their faithfulness to the decision-making process of models (Adebayo et al., 2018).

While our approach shares similarities with these methods, it leverages the intrinsic properties of ViTs and employs a more principled strategy using token discarding.

**Datasets for investigating spurious correlations.** The majority of research investigating spurious correlations after training focuses on datasets with human-identifiable features, such as the CelebA dataset (Liu et al., 2015), where celebrities are categorized by characteristics like hair color and gender. As a result, many research efforts examine spurious correlations as a group-level phenomenon, using methods such as worst-case generalization of group performance (Sagawa et al., 2019; Ghosal & Li, 2024). Unfortunately, this approach makes the proposed methods largely inapplicable to datasets that lack features that are easily identifiable by humans.

**Occlusion-based methods to detect spurious regions.** A common strategy in interpretability is to explain model behavior by removing parts of the input and observing how predictions change. Early work on representation erasure demonstrated that selectively removing features can reveal which components a model relies on for its decisions (Li et al., 2016). This idea was later unified under a "removal-based explanation" perspective, which characterizes many attribution methods as instances of simulating feature removal and measuring the resulting change in model output (Covert et al., 2021). In practice, feature removal in interpretability is typically implemented through replacement strategies. A feature can either be marginalized by replacing it with sampled plausible values from a conditional distribution of the input (Lundberg & Lee, 2017; Zintgraf et al., 2017; Frye et al., 2020), or replaced with a fixed baseline value for occlusion. Common choices for occlusion include zeroing the feature (Zeiler & Fergus, 2014; Petsiuk et al., 2018; Schwab & Karlen, 2019), replacing it with a mean or adjacent value (Zhou et al., 2014; Ribeiro et al., 2016; Dabkowski & Gal, 2017), or using blurred versions of the input (Fong & Vedaldi, 2017).

Our approach follows the same removal-based perspective but differs in that we remove tokens directly from the input sequence during inference, rather than substituting them with predefined baseline values. Leveraging the tokenized representation unique to ViTs, we assess token influence without introducing artificial replacement values.

**Token discarding in ViTs.** Token discarding is made possible by a property unique to transformer-based architectures, where tokens can be explicitly removed from the input sequence without requiring spatial interpolation or feature replacement (Rao et al., 2021). Beyond interpretability (Haurum et al., 2023; Pan et al., 2021), token discarding has been explored for improving model robustness (Renggli et al., 2022), accelerating training and inference (Bolya et al., 2022; Rao et al., 2021; Long et al., 2023).

In this work, we adopt token discarding specifically as a principled mechanism for estimating token influence, which allows us to quantify the change in prediction confidence when individual tokens are removed from the full token set.

## 3 Methodology

### 3.1 Identifying spurious correlations via token discarding

We propose a two-step procedure to identify spurious correlations in ViTs. Given an image and its corresponding bounding box, we:

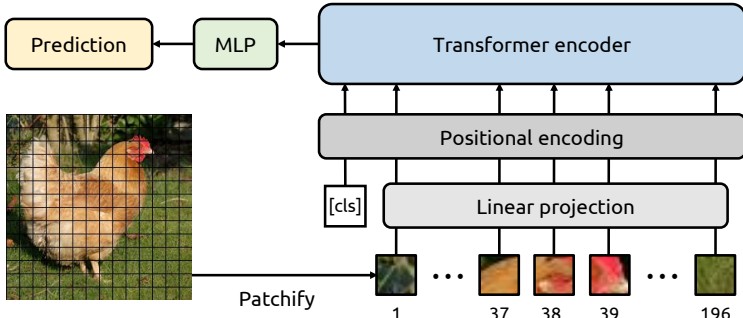

Figure 2: An overview of the ViT architecture and the tokenization of image patches.

- **Step 1**: Discover influential tokens that contribute to the prediction made by the model.

- **Step 2**: Identify spurious correlations based on influential tokens and bounding box information.

### 3.1.1 Discovering influential tokens

**Notation.** Given a correctly classified image $\mathtt{X}$ and a ViT with parameters $\theta$, we denote the corresponding prediction confidence for the ground-truth class label ($c$) of that image as $\hat{y} = g_\theta(\mathtt{X})_c$, where $g_\theta(\cdot)$ describes a forward pass, mapping inputs to output predictions. In our setup, ViT-B/16 tokenizes an input image $\mathtt{X}$ of size $224 \times 224$ into 196 tokens which we will denote as $\mathtt{X} = [\boldsymbol{x}_i]_{i \in \{1,\ldots,196\}}$.

**Finding influential tokens.** To identify the influential tokens, we systematically remove one token at a time from the image and observe how each removal affects the model's prediction compared to its initial prediction. Formally, we define our token discarding approach as $\mathtt{X}_{-k} = [\boldsymbol{x}_i]_{i \in \{1,\ldots,196\}\setminus\{k\}}$, where the $k$-th token has been discarded from the input image. The prediction for the newly created input with the discarded token $k$ then becomes $\hat{y}_{-k} = g_\theta(\mathtt{X}_{-k})_c$.

**Definition 1** (Token Influence Map). *For each token index $k \in \{1,\ldots,196\}$, let $\mathtt{X}_{-k} = [\boldsymbol{x}_i]_{i \in \{1,\ldots,196\}\setminus\{k\}}$ denote the input obtained by discarding the $k$-th token, and let $\hat{y}_{-k} = g_\theta(\mathtt{X}_{-k})_c$ be the corresponding prediction confidence for the ground-truth class $c$. The* influence *of token $k$ is defined as*

$$\Delta_k = \hat{y} - \hat{y}_{-k}.$$

*The vector*

$$\mathtt{z} = [\Delta_k]_{k=1}^{196}$$

*is called the* token influence map.

**Leave-one-out (LOO) attribution.** LOO methods estimate feature importance by measuring the change in a model's prediction when a single feature is removed from the input. This strategy has been widely adopted in machine learning to assess feature and context importance through single-feature ablation (Zeiler & Fergus, 2014; Li et al., 2016; Ribeiro et al., 2016; Cohen-Wang et al., 2024). Our method follows a leave-one-out removal strategy that measures the marginal effect of removing a single token from the full input.

From a cooperative game-theoretic perspective, this quantity corresponds to the marginal contribution of token $k$ with respect to the grand coalition of all tokens. In particular, up to a constant scaling factor, the influence of token $k$ corresponds to the first-order contribution (main effect) in the Shapley value $\phi_k$:

$$\phi_k = \frac{1}{196} \Delta_k + (\text{H.O.}),$$

where the higher-order terms (H.O.) capture interaction effects arising from coalitions involving two or more tokens. Alternatively, our method can be viewed as fitting a weighted linear surrogate model to the prediction

function in which all weight is concentrated on coalitions that differ from the full input by at most one token (see Lundberg & Lee, 2017, Appendix B.2).

**Occlusion and token discarding.** LOO attribution is commonly implemented through occlusion, where a region of the input image is masked or replaced with a predefined value (e.g. zero, gray, or blur), and the resulting change in prediction is used as an importance score. Our method follows the same leave-one-out principle, but operates at the level of vision transformer tokens. However, instead of occluding a token by value replacement, we remove the token entirely from the input sequence and re-evaluate the prediction. As a result, our method implements feature absence without replacement, which avoids occlusion patterns and directly measures each token's contribution to the model's prediction.

**Inherited properties.** As our method implements a standard LOO intervention, the resulting token influence scores inherit several well-known properties of LOO attribution (Lundberg & Lee, 2017; Sundararajan & Najmi, 2020; Covert et al., 2021): (i) *Dummy* – tokens whose removal does not change the prediction receive zero influence; (ii) *Consistency* – each score is deterministic and uniquely defined for a given input and model; (iii) *Monotonicity* – scores induce a total ordering over tokens reflecting their relative contribution; (iv) *Counterfactual faithfulness* – each score is derived from a single, well-defined counterfactual input. Additionally, we note that the influence maps are intentionally *model-sensitive*: two models with different parameters producing the same prediction may yield different influence maps. This behavior is intentional, as the goal of our method is to reveal differences in how models utilize input tokens, thus identifying spurious correlations.

**Missingness bias.** Occlusion-based attribution methods simulate feature absence through value replacement, which can induce out-of-distribution inputs and cause prediction changes that are unrelated to true information removal (Jain et al., 2022; Balasubramanian & Feizi, 2023). This phenomenon, known as *missingness bias*, arises when the model responds to the artificial replacement itself rather than to the absence of the feature.

Our method discards tokens entirely, representing feature absence through structural token removal instead of value substitution. As a result, the perturbed input remains within the model's natural input space, and the resulting prediction difference isolates the effect of information removal from masking-induced artifacts. Here, token removal in ViTs implements true feature absence and thereby mitigates missingness bias, as demonstrated by Jain et al. (2022). Since our approach directly builds on this finding and relies exclusively on token removal, it is robust to such biases. We empirically demonstrate this property in Appendix B.

**Consistency across models.** Unlike Shapley-based attribution methods, which aim to produce consistent feature importance scores across models with similar behavior, our method does not enforce model-level consistency.

### 3.1.2 Identifying spurious correlations

Given an image with bounding box information, we first identify the tokens that lie within the bounding box as shown in Figure 3. Based on this information, we create two sets of token IDs, one for those that lie in the bounding box ($B_{in}$) and one for those that are outside it ($B_{out}$). Afterwards, using the previously created token influence maps, we are able to quantify whether the identified influential tokens lie within the bounding box or not. We propose the following two metrics to identify spurious correlations:

- **Average Token Spuriosity Index (A-TSI)**. The first metric we propose is based on the average token influence and is described as follows:

$$\text{A-TSI}(Z, B_{in}, B_{out}) = \frac{\frac{1}{n_{out}} \sum_{i \in B_{out}} z_i}{\frac{1}{n_{in}} \sum_{i \in B_{in}} z_i} . \tag{1}$$

- **Maximum Token Spuriosity Index (M-TSI)**. The second metric we propose is based on the maximum token influence and is described as follows:

$$\text{M-TSI}(Z, B_{in}, B_{out}) = \frac{\max(\{z_i\}_{i \in B_{out}})}{\max(\{z_i\}_{i \in B_{in}})} . \tag{2}$$

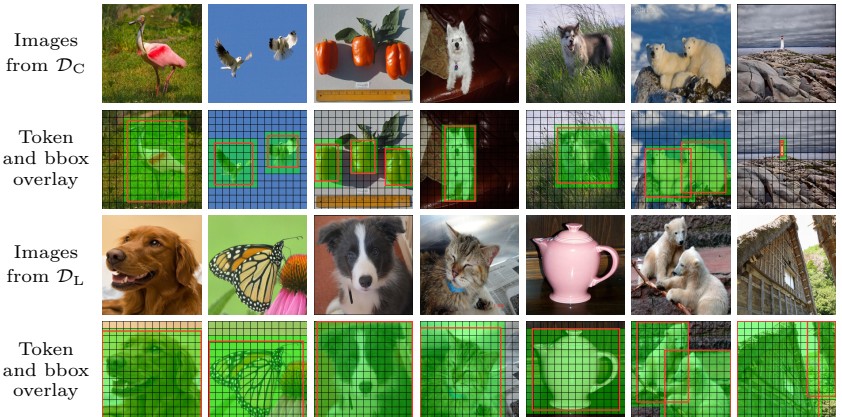

Figure 3: Example images from the subset of images correctly classified by all models ($\mathcal{D}_\mathrm{C}$) and the subset of images with bounding boxes covering almost the entire image ($\mathcal{D}_\mathrm{L}$) are provided, together with their bounding boxes (red) and corresponding tokens based on $16 \times 16$ patches (green). As can be seen, bounding boxes for images from $\mathcal{D}_\mathrm{L}$ cover almost the entirety of the image.

While both A-TSI and M-TSI are useful in identifying spurious correlations, A-TSI measures average spurious correlation influence relative to core features, while M-TSI targets scenarios where a few influential tokens outside the bounding box signal significant correlations.

**Interpreting TSI scores**. Both A-TSI and M-TSI are straightforward to understand in terms of what their scores indicate.

- TSI $\in (0, 1)$ implies that the tokens within the bounding box are more influential for the prediction compared to those that are outside, meaning that the prediction is based on features that originate from regions of the input image related to the class information.

- TSI $= 1$ indicates that features outside the bounding box are as important as those that are inside for prediction.

- TSI $> 1$ implies the influence of tokens outside is greater than those inside, indicating the existence of spurious correlations. As TSI increases beyond 1, the intensity of spurious correlations increases.

In order to provide a straightforward understanding of the proposed method, we present Figure 4, which displays input images and their corresponding token influence maps overlaid with bounding box information. For each image, we calculate M-TSI and A-TSI, sorting the images from left to right according to increasing M-TSI scores. As measured by TSI, images on the left side display lower levels of spurious correlation since most of the important tokens are inside the bounding box, whereas images on the right side demonstrate higher levels of spurious correlation since the important tokens are outside the bounding box, indicating that the model is relying on irrelevant features or background information for its predictions.

**Limitations of the bounding-box assumption.** Both A-TSI and M-TSI assume that the bounding box delineates the core features relevant to classification. However, as we discuss in Section 4 (see Figure 8), this assumption can break down in the presence of label inconsistencies, secondary objects, or extremely small areas of interest. In such cases, TSI scores should be interpreted as measuring a model's reliance on features outside the annotated region, which serves as a proxy for, but not identical to, reliance on genuinely spurious features.

## 3.2 Dataset

We conduct our experiments on the ImageNet validation set (Russakovsky et al., 2015), which is the most widely used large-scale image classification benchmark and is therefore well suited for studying spurious

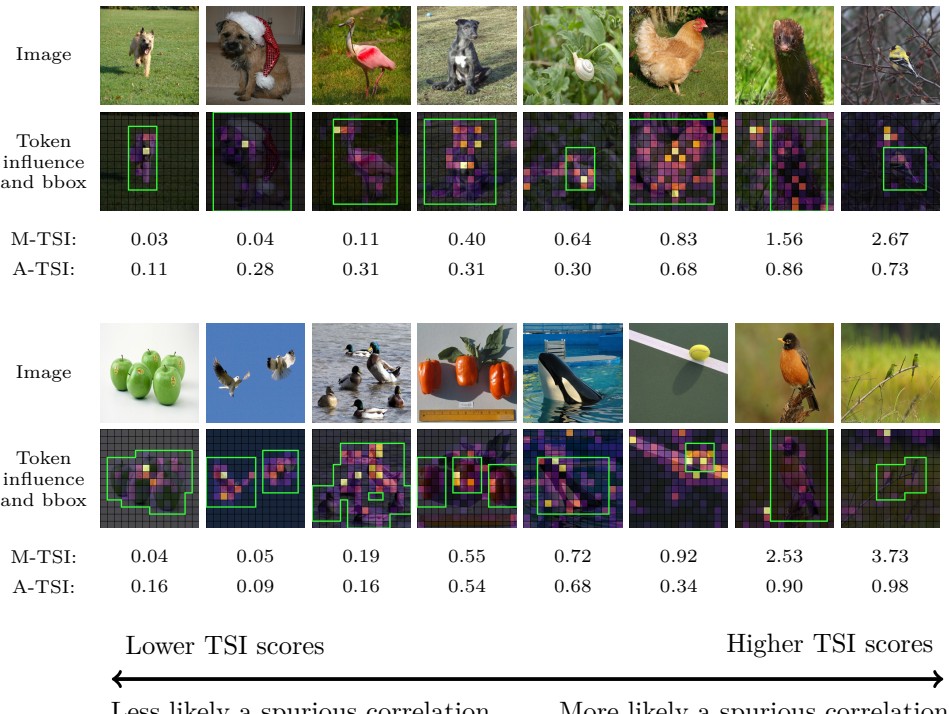

Figure 4: An example set of images is presented with their corresponding token influence maps generated using ViTs, as well as M-TSI and A-TSI scores. Green boundaries represent the bounding box information, highlighting the object of interest in the image. The images are sorted according to an increasing value of M-TSI to provide a clear qualitative view of spurious correlations identified by the proposed method.

correlations at scale. This dataset contains approximately 50,000 images spread over 1,000 classes, all of which come with human-annotated bounding boxes highlighting the object of interest in each image, provided as part of the ILSVRC localization task (Russakovsky et al., 2015).

Instead of performing experiments at an aggregate level on all images in the ImageNet validation set, we separate them into three groups based on the properties detailed in the following.

- **Bounding box for the object of interest**. Recall that the images in the ImageNet dataset are tokenized into 196 tokens by ViTs. Surprisingly, we discovered that a large subset of images in this dataset have bounding boxes covering almost the entire image (over 160 tokens); we denote this subset as $\mathcal{D}_L$. Some examples of images that exhibit this phenomenon are shown in Figure 3. For a faithful analysis, we filter out those $11,221$ images and do not use them in our experiments. This criterion ensures a targeted evaluation of the degree of spurious correlation by placing emphasis on images where the object of interest does not constitute a significant portion of the image.

- **Classification accuracy**. Inspired by the work of Ozbulak et al. (2021a), we divide the remaining images into two categories: those that are classified correctly by all selected models and those that are incorrectly classified by at least one model. This division ensures that our experiments capture differences between images that are easy to classify (i.e., those that are correctly classified by all models) and those that are comparatively harder to classify.

  - **Images that are correctly classified by all models ($\mathcal{D}_C$)**. This filtering operation ensures that the selected set of images has a consensus among all models in their classification, allowing for a more controlled evaluation across models. Based on this criterion, we find that $26,317$ images are correctly classified by all three selected models.

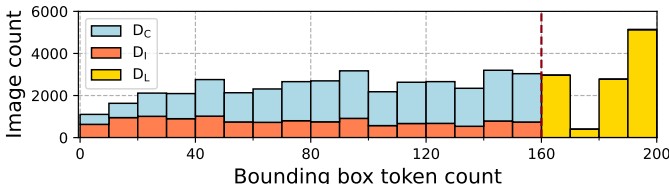

Figure 5: Stacked histograms showing the distribution of images in the ImageNet validation dataset based on the coverage of bounding boxes, illustrated in relation to the number of tokens. The dataset is grouped into images with bounding boxes covering almost the entire image ($\mathcal{D}_\mathrm{L}$), and, from the remaining subset, images correctly classified by all models ($\mathcal{D}_\mathrm{C}$) and images misclassified by at least one model ($\mathcal{D}_\mathrm{I}$).

- **Images that are incorrectly classified by at least one model ($\mathcal{D}_\mathbf{I}$)**. To explore potential relationships between spurious correlations and classification accuracy, we conduct a separate analysis for images that are incorrectly classified by at least one model. By isolating these images, our idea is to gain insight into the specific challenges that arise in the classification process and spurious correlations. This category comprises $12,458$ images.

Based on the grouping detailed above, we can represent the ImageNet validation dataset as a combination of three disjoint subsets $\mathcal{D}_\mathrm{ImageNet} = \mathcal{D}_\mathrm{L} \cup \mathcal{D}_\mathrm{C} \cup \mathcal{D}_\mathrm{I}$ (see Figure 5). For experiments, we will use $\mathcal{D}_\mathrm{C}$ and $\mathcal{D}_\mathrm{I}$.

### 3.3 Models

We employ the most commonly used Vision Transformer architecture: ViT-B/16 (Dosovitskiy et al., 2020). The ViT-B/16 architecture tokenizes the input image into patches of size $16 \times 16$ (see Figure 2), resulting in 196 tokens for a standard ImageNet image of size $224 \times 224$.

In addition to the model trained in a supervised fashion, we also use two additional models based on the same architecture but trained in a self-supervised manner: Self-Distillation with No Labels (DINO) (Caron et al., 2021) and Masked AutoEncoders (MAE) (He et al., 2022). DINO employs a novel self-supervised learning approach that relies on contrastive learning whereas MAE further advances this concept by learning to reconstruct missing parts of input images, effectively capturing intricate details and dependencies within the data. Further details on the employed self-supervised training methods can be found in their respective papers and in the surveys of Khan et al. (2022) and Ozbulak et al. (2023).

## 4 Experimental results and insights on spurious correlations

Using all images in $\mathcal{D}_C$ and $\mathcal{D}_I$ and with three ViTs, we calculate A-TSI and M-TSI with token influence maps and present the mean and standard deviation in Table 1. Apart from providing these parameters for the group of all images, we also filter images according to the size of the bounding box in terms of token coverage and calculate the aforementioned metrics over four groups: 1-40, 41-80, 81-120, and 121-160 tokens. We also provide extensive histograms of TSI in the Appendix (Figure 11) for $\mathcal{D}_C$ and $\mathcal{D}_I$. Based on these results, we can make the following observations.

**Correctly classified images contain fewer spurious features**. A-TSI and M-TSI values for correctly classified images are generally lower across all token coverage groups, suggesting that models focus more on relevant features when making accurate predictions. Conversely, misclassified images tend to have higher TSI values, indicating the presence of more spurious or irrelevant features that may mislead the model. Notably, the M-TSI in the $[2, 2+]$ bin in Figure 11 highlights that a large number of images are incorrectly classified by at least one model, likely due to strong spurious correlations as identified by M-TSI. In contrast, this bin contains proportionally fewer images in $\mathcal{D}_C$, despite the fact that $\mathcal{D}_C$ has a larger overall number of images compared to $\mathcal{D}_I$.

**The training method influences spurious correlations**. Although the three selected models share the same architecture, we observe differing results when analyzing TSI. Specifically, for images in $\mathcal{D}_C$, DINO

Table 1: Using ViTs trained with supervised learning, DINO, and MAE, mean (standard deviation) TSI is calculated for $\mathcal{D}_C$ (i.e., images that are correctly classified) and $\mathcal{D}_I$ (i.e., images that are misclassified) for all images within subsets as well as certain groups of images based on the coverage of the bounding box in terms of tokens.

| $\mathcal{D}_C$ | Tokens | Supervised | DINO | MAE | $\mathcal{D}_I$ | Tokens | Supervised | DINO | MAE |
|---|---|---|---|---|---|---|---|---|---|
| **M-TSI** | All | 0.64 (0.76) | 0.35 (0.64) | 0.68 (0.69) | **M-TSI** | All | 0.94 (1.64) | 0.83 (1.30) | 0.84 (1.38) |
| | 1-40 | 0.88 (1.22) | 0.64 (1.44) | 0.86 (1.19) | | 1-40 | 1.63 (2.47) | 1.47 (1.82) | 1.38 (1.96) |
| | 41-80 | 0.68 (0.73) | 0.37 (0.43) | 0.73 (0.63) | | 41-80 | 0.85 (1.13) | 0.76 (1.14) | 0.81 (1.14) |
| | 81-120 | 0.62 (0.67) | 0.30 (0.33) | 0.67 (0.55) | | 81-120 | 0.57 (0.70) | 0.45 (0.52) | 0.53 (0.56) |
| | 121-160 | 0.51 (0.53) | 0.24 (0.26) | 0.57 (0.47) | | 121-160 | 0.39 (0.42) | 0.32 (0.39) | 0.40 (0.76) |
| **A-TSI** | All | 0.54 (0.30) | 0.33 (0.19) | 0.61 (0.27) | **A-TSI** | All | 0.51 (0.34) | 0.44 (0.30) | 0.49 (0.29) |
| | 1-40 | 0.37 (0.28) | 0.26 (0.23) | 0.41 (0.25) | | 1-40 | 0.51 (0.40) | 0.47 (0.34) | 0.46 (0.33) |
| | 41-80 | 0.47 (0.24) | 0.30 (0.16) | 0.56 (0.23) | | 41-80 | 0.50 (0.32) | 0.45 (0.32) | 0.50 (0.30) |
| | 81-120 | 0.57 (0.28) | 0.34 (0.17) | 0.65 (0.24) | | 81-120 | 0.50 (0.31) | 0.42 (0.28) | 0.51 (0.26) |
| | 121-160 | 0.64 (0.32) | 0.37 (0.18) | 0.71 (0.27) | | 121-160 | 0.52 (0.29) | 0.41 (0.24) | 0.52 (0.26) |

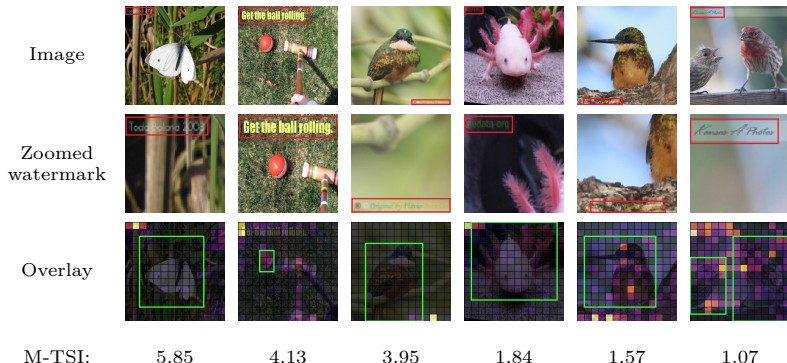

Image

Zoomed watermark

Overlay

M-TSI:   5.85   4.13   3.95   1.84   1.57   1.07

Figure 6: Example images obtained from the ImageNet dataset where tokens highlight watermarks over the image instead of the objects of interest, leading to high TSI scores.

yields lower TSI scores than the other two models, suggesting that the training method influences how strongly models rely on features outside the annotated region. Notably, while DINO has the lowest TSI among the three, MAE shows the highest despite its strong predictive performance (see Table 1). This observation is consistent with prior findings that high predictive performance does not guarantee reliance on robust or semantically meaningful features (Geirhos et al., 2020), and that models with comparable accuracy can differ substantially in their dependence on core features (Singla et al., 2022). In contrast, for images in $\mathcal{D}_I$, the TSI values of the different models are less distinct, with DINO again having the lowest median TSI by a small margin. Consequently, DINO appears to place greater emphasis on core features, suggesting that it may be the more robust ViT. Supporting the claims made by Caron et al. (2021), we believe the robustness of DINO stems from its distillation-based training routine which mostly transfers robust and useful features between teacher and student networks.

**M-TSI captures spurious correlations more effectively for small objects**. While A-TSI remains consistent across images with varying bounding box sizes, M-TSI differs significantly, particularly for images containing relatively small objects of interest. Specifically, images with bounding boxes covering up to 40 tokens show higher M-TSI values compared to other groups and their corresponding A-TSI. This suggests that when the object of interest is relatively small, the M-TSI score is more effective in detecting spurious correlations than the A-TSI score.

**M-TSI identifies strong spurious signals such as watermarks**. In Figure 6, we provide another set of qualitative examples where the tokens covering the watermark are identified as influential tokens, leading to high TSI scores. This discovery reveals yet another use case for the proposed method where spurious correlations based on watermarks can be identified using M-TSI. This capability demonstrates the

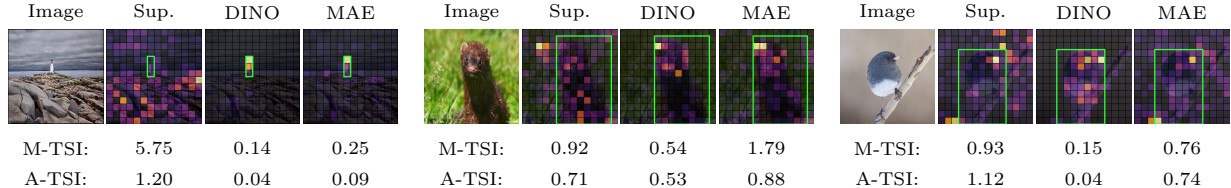

| Image | Sup. | DINO | MAE | Image | Sup. | DINO | MAE | Image | Sup. | DINO | MAE |
|---|---|---|---|---|---|---|---|---|---|---|---|
| M-TSI: | 5.75 | 0.14 | 0.25 | M-TSI: | 0.92 | 0.54 | 1.79 | M-TSI: | 0.93 | 0.15 | 0.76 |
| A-TSI: | 1.20 | 0.04 | 0.09 | A-TSI: | 0.71 | 0.53 | 0.88 | A-TSI: | 1.12 | 0.04 | 0.74 |

Figure 7: Example images in the ImageNet dataset, where the important tokens are correctly identified by DINO and MAE, resulting in low TSI scores, whereas the supervised model fails to do so, leading to high TSI scores.

Table 2: For images in $\mathcal{D}_I$ (i.e., images misclassified into incorrect categories), M-TSI and A-TSI are computed and grouped into one of four bins based on prediction confidence. For each bin, the mean (std) TSI values are calculated.

| $\mathcal{D}_I$ | Model | 0-25% | 25-50% | 50-75% | 75-100% |
|---|---|---|---|---|---|
| **M-TSI** | Supervised | 1.13 (1.44) | 0.83 (1.22) | 0.77 (1.27) | 0.87 (1.91) |
| | DINO | 0.98 (1.19) | 0.72 (0.96) | 0.64 (1.00) | 0.79 (1.51) |
| | MAE | 1.07 (1.67) | 0.76 (1.00) | 0.73 (1.29) | 0.81 (1.39) |
| **A-TSI** | Supervised | 0.54 (0.37) | 0.48 (0.32) | 0.47 (0.32) | 0.50 (0.32) |
| | DINO | 0.47 (0.32) | 0.43 (0.29) | 0.41 (0.28) | 0.42 (0.28) |
| | MAE | 0.50 (0.36) | 0.47 (0.30) | 0.46 (0.26) | 0.53 (0.28) |

Table 3: Classes with the highest average M-TSI scores in the ImageNet validation dataset calculated for $\mathcal{D}_C$ (i.e., images that are correctly classified) and $\mathcal{D}_I$ (i.e., images that are misclassified). Repeating classes are highlighted in bold.

| Image subset | Supervised | | DINO | | MAE | |
|---|---|---|---|---|---|---|
| | Class | M-TSI | Class | M-TSI | Class | M-TSI |
| $\mathcal{D}_C$ | **space bar** | 4.99 | **ping-pong ball** | 4.83 | **ping-pong ball** | 3.18 |
| | **puck** | 3.84 | **space bar** | 3.69 | **puck** | 3.09 |
| | **ping-pong ball** | 3.35 | **puck** | 3.26 | **space bar** | 2.56 |
| | **geyser** | 1.90 | **alp** | 2.03 | **rapeseed** | 1.68 |
| | **rugby ball** | 1.87 | balance beam | 1.55 | **basketball** | 1.68 |
| | laptop | 1.78 | **volleyball** | 1.55 | **volleyball** | 1.68 |
| | **basketball** | 1.76 | diaper | 1.47 | miniskirt | 1.52 |
| | shoji | 1.76 | **rapeseed** | 1.45 | apiary | 1.51 |
| | **rapeseed** | 1.75 | pickelhaube | 1.37 | **alp** | 1.48 |
| | lakeside | 1.69 | **rugby ball** | 1.30 | **geyser** | 1.48 |
| $\mathcal{D}_I$ | shoe shop | 7.06 | **cockatoo** | 7.17 | **cockatoo** | 7.73 |
| | **worm fence** | 7.00 | **soccer ball** | 5.35 | parking meter | 5.07 |
| | airship | 6.77 | conch | 4.75 | radiator | 4.46 |
| | Granny Smith | 5.80 | traffic light | 4.03 | matchstick | 3.97 |
| | flagpole | 5.33 | baseball | 3.89 | cockroach | 3.88 |
| | bow tie | 4.90 | **worm fence** | 3.65 | brass | 3.67 |
| | **soccer ball** | 4.85 | pickelhaube | 3.30 | jean | 3.26 |
| | **platypus** | 3.84 | tennis ball | 3.23 | **platypus** | 3.21 |
| | slug | 3.60 | crash helmet | 3.03 | cello | 3.06 |
| | zebra | 3.56 | accordion | 2.99 | grille | 3.06 |

versatility and robustness of M-TSI in uncovering subtle, yet impactful, spurious correlations that could compromise the model's performance in real-world scenarios.

**Training routine affects TSI on identical images**. We present Figure 7, which shows the TSI calculated using three ViT models for identical images. In the provided examples, the supervised model considers spurious features that surround the object for the prediction, while DINO and MAE successfully make use of the features originating from the class-related regions of the image. This shows that the features taken into account during prediction may vary by the type of learning, resulting in different degrees of spuriosity for different models on the same dataset. We further explore model differences in the next section.

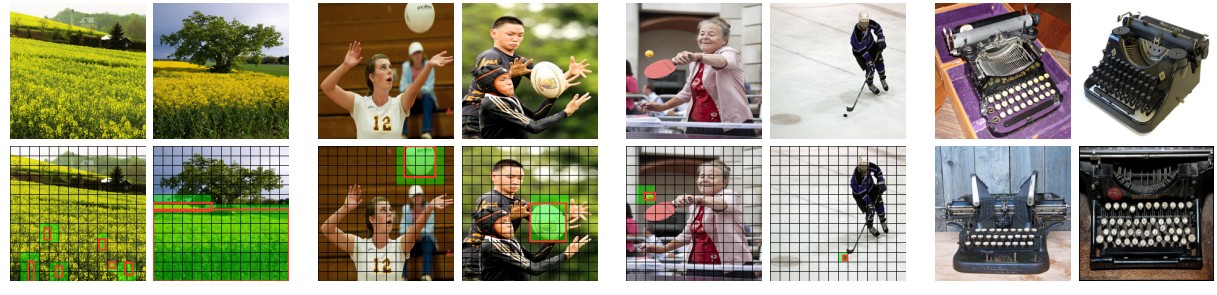

(a) Label inconsistency     (b) Secondary objects     (c) Small area of interest     (d) Class similarities

Figure 8: Several examples from the ImageNet dataset and their annotations, highlighting (a) label inconsistencies, (b) the presence of secondary objects, (c) small areas of interest, and (d) class similarities, all of which contribute to higher TSI scores. In (a), two images originate from the same class (rapeseed), but in the first image, the annotation covers only a few individual plants, whereas in the second image, it encompasses the entire field. In (b), the presence of humans, along with other strong correlating signals such as uniforms or helmets, influences the focus of models. In (c), the objects of interest are extremely small, making it difficult for models to detect them. In (d), four images come from two distinct classes: typewriter keyboard and spacebar, despite the images and their backgrounds being nearly identical.

**Misclassifications with low confidence have higher TSI scores**. Investigating TSI for misclassified images, we discover that misclassifications with low confidence (i.e., confidence between 0% and 25%) have substantially higher M-TSI and A-TSI scores, indicating that when the model is uncertain about its prediction, it is more likely to rely on non-core features rather than the primary object of interest (see Table 2). This trend is particularly pronounced for M-TSI, indicating that, in such cases, the most influential tokens for the prediction often reside outside the bounding box of the object, highlighting the model's tendency to use background artifacts or incidental cues when making uncertain decisions.

**Class-based investigation with TSI scores**. In order to investigate whether some classes are more prone to spurious correlations than others, we calculate the TSI for images within each class separately. In Table 3, we provide the 10 classes with the highest average M-TSI scores for each model along with their respective values for images in $\mathcal{D}_C$ and in $\mathcal{D}_I$. Notably, a large portion of the classes repeatedly appears across all three models, suggesting the presence of potentially systematic problems in the ImageNet dataset specific to these classes. This repeated appearance raises concerns about biases or artifacts that may be embedded within the dataset, leading models to rely on irrelevant features when making predictions. We also notice that classes in $\mathcal{D}_I$ have substantially higher M-TSI scores compared to classes in $\mathcal{D}_C$, meaning that misclassifications often rely on unintended cues, rather than the features of the object of interest.

To uncover the underlying reasons for the presence of spurious correlations, we investigate images with high TSI scores, particularly M-TSI scores. After analyzing several hundred images, along with their token influence maps and TSI scores, we identify the following key factors contributing to spurious correlations:

- Label inconsistency: In these cases, the bounding box either fails to highlight the correct object of interest, partially covers it, or includes unrelated parts of the image. Such inconsistencies can mislead models into associating predictions with irrelevant features.

- Secondary objects: Some images contain secondary objects that strongly correlate with the primary object of interest, leading models to focus on these unintended features. For example, several ImageNet classes related to sports frequently include humans in the images. In such cases, we find that models tend to focus on the person rather than the actual object of interest.

- Small area of interest: Some target objects in the dataset are extremely small, often occupying only a single token in the model's representation. Examples include ping-pong, tennis, and rugby balls, which appear in images but are too small for the model to reliably detect. Instead of focusing on the actual object, the model often relies on contextual cues from the rest of the image to infer the class, leading to high TSI.

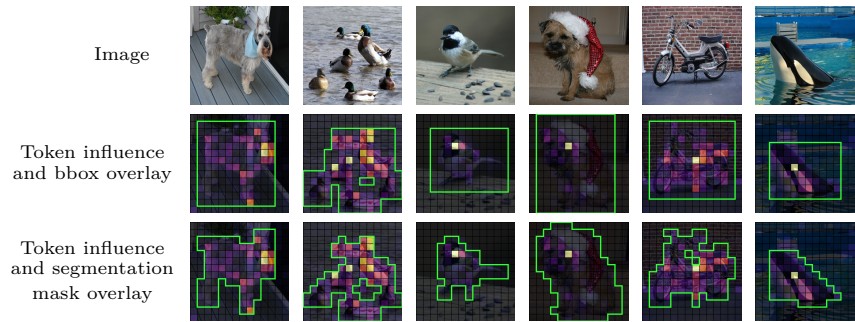

Figure 9: Example images from the ImageNet dataset (top) and their corresponding token influence maps with selected tokens highlighting the object of interest using bounding box annotations (middle) and segmentation masks (bottom).

- Class similarities: Certain classes exhibit significant visual similarities, making it difficult for the model to distinguish between them. For instance, the classes typewriter, computer keyboard, and typewriter keyboard are different classes in ImageNet but contain highly similar objects and backgrounds. This overlap can cause models to rely on features that are not unique to a single class, reducing their ability to generalize correctly.

For each of the aforementioned cases, we provide several examples and their descriptions in Figure 8. We also provide a detailed report for all classes in ImageNet, similar to Table 3, as supplementary material, highlighting classes that rely on spurious correlations based on M-TSI. For class-based analysis, our results can be investigated together with reports provided by Singla & Feizi (2021), whose work also explores spurious correlations at the class level. Note that our work is not the only research effort that identified these problems in the ImageNet dataset (Ozbulak et al., 2021b; Peychev et al., 2023; Beyer et al., 2020; Moayeri et al., 2022; Singla & Feizi, 2021). However, our method can consistently discover such cases based on TSI scores during post-hoc analysis of trained models.

**Using segmentation masks instead of bounding boxes**.

In addition to bounding box annotations, we conduct experiments using semantic segmentation masks to examine whether annotation granularity affects the resulting TSI scores. For segmentation masks, we use the annotations provided by Gao et al. (2022) for 12,419 images from the ImageNet validation set, of which 7,117 belong to $\mathcal{D}_C$.

For this subset, we compute TSI scores using tokens selected by (1) bounding box annotations and (2) segmentation masks, and compare the resulting A-TSI and M-TSI values. In most cases, the difference between segmentation- and bounding box–based TSI scores is less than 0.05, with M-TSI scores generally increasing and A-TSI scores decreasing due to tighter boundaries (see Figure 9). However, we find that noticeable differences arise in some instances, typically when bounding boxes cover substantially larger regions than the corresponding segmentation masks (e.g., for diagonally oriented or sparsely shaped objects), leading to larger TSI deviations. We provide further details and additional results on this topic in Appendix D.

## 5 Case study on breast cancer detection

Based on our previous findings, we tackle a real-world scenario involving spurious correlations: invasive breast mass classification on mammography images. For this case study, we use the VinDr-Mammo dataset, which is designed to advance computer-aided diagnosis in full-field digital mammography (Nguyen et al., 2023). We fine-tune three pretrained ViTs on this dataset, making sure that the models achieve performance close to the state-of-the-art on this dataset. Based on the trained models, we calculate TSI scores based on the breast area and investigate images with their corresponding TSI scores and present several cases in Figure 10.

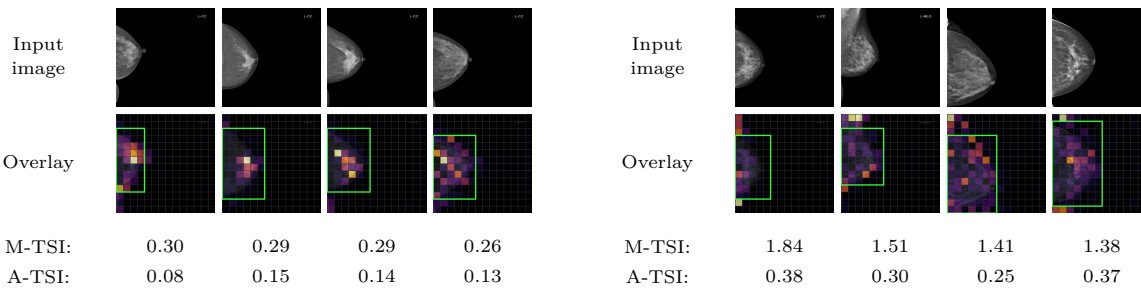

| | | | | | | | | | |
|---|---|---|---|---|---|---|---|---|---|
| M-TSI: | 0.30 | 0.29 | 0.29 | 0.26 | M-TSI: | 1.84 | 1.51 | 1.41 | 1.38 |
| A-TSI: | 0.08 | 0.15 | 0.14 | 0.13 | A-TSI: | 0.38 | 0.30 | 0.25 | 0.37 |

(a) Examples without spurious correlations
(b) Examples with spurious correlations

Figure 10: Images from the VinDr-Mammo dataset (top) and their corresponding token influence maps obtained from fine-tuned models (bottom). The left side presents examples with lower M-TSI and A-TSI scores, indicating the absence of spurious correlations. The right side shows examples with higher M-TSI and A-TSI scores, where spurious correlations are evident in regions outside the breast area, such as chest fat tissue.

As illustrated by the images on the left side of Figure 10, where M-TSI and A-TSI scores are lower, the models concentrate on the pertinent breast tissue areas. In contrast, for images on the right side, which have higher M-TSI and A-TSI scores, the models assign importance to non-relevant regions outside the breast, such as chest fat tissue. These examples demonstrate the potential of our proposed method in effectively identifying spurious correlations in real-world scenarios, ensuring that model predictions are based on correct features (i.e., clinically meaningful features) rather than on irrelevant artifacts.

# 6 Selecting A-TSI vs M-TSI

Throughout this paper, we presented results and examples containing both M-TSI and A-TSI. However, a fundamental question arises: given a model trained on a dataset, which of the two metrics is more appropriate to identify spurious correlations? Based on previous observations, we investigate whether a large A-TSI is indicative of a large M-TSI, and vice versa.

Given the differences between A-TSI and M-TSI, it is important to determine which measure is more useful for identifying spurious correlations. Across all experiments that use token influence maps, we found that M-TSI provides a more consistent and reliable signal for detecting spurious correlations, particularly in cases where the relevant areas of the object of interest cover large portions of the image. In such cases, A-TSI tends to smooth out due to the large number of tokens covering the area, whereas M-TSI, which is calculated using only the maximum token influence, more reliably identifies spurious correlations. Therefore, based on our findings, M-TSI emerges as the more appropriate metric for evaluating the extent to which a model relies on spurious features under ideal conditions, making it a preferable choice for assessing model trustworthiness in real-world applications.

# 7 Conclusions and future perspectives

In this work, we proposed an easily deployable token-based technique, together with two complementary metrics, to identify spurious correlations in ViTs. Through large-scale experiments on ImageNet, we demonstrated that the chosen training routine can have a substantial impact on a model's reliance on non-core visual cues, even when the underlying architecture is fixed. Beyond aggregate statistics, our analysis enabled fine-grained, image-level inspection of spurious behavior, revealing recurring dataset artifacts such as watermarks, secondary objects, and annotation inconsistencies. We further conducted class-level investigations to identify categories that consistently exhibit strong spurious signals across models, and examined how factors such as object size and annotation granularity influence spuriosity estimates. Finally, we validated the practical relevance of the proposed metrics through additional analyses using segmentation masks and a

real-world medical imaging case study, illustrating the applicability of token-level spuriosity analysis beyond standard image classification benchmarks.

Note that our evaluation relies on bounding-box or segmentation-mask annotations as proxies for core feature regions, and that the token influence scores and TSI have not been validated against ground-truth spurious correlation labels, which are rarely available at scale. To partially address this limitation, we provide additional ablations and comparisons related to TSI in Appendix C, offering further empirical insight into its behavior across settings.

While this work focuses on post-hoc identification of spurious correlations in vision transformers, several promising research directions remain open. A natural extension is to move from detection to intervention, where token-level spuriosity signals could be used to actively guide training, for example by regularizing models to down-weight consistently spurious tokens or by selectively reweighting samples exhibiting high TSI. Another important direction is to relax the reliance on explicit spatial annotations by exploring weakly supervised or annotation-free alternatives. Beyond image classification, extending the proposed framework to dense prediction tasks (e.g., detection, segmentation) and multimodal settings would allow the study of spurious correlations in more realistic and safety-critical pipelines. Finally, the token-based nature of our method opens the door to theoretical analysis of spurious correlations in transformer representations, including connections to token redundancy, representation collapse, and robustness under distribution shift. We believe these directions position token-level spuriosity analysis as a useful building block for both understanding and mitigating shortcut learning in modern vision models.

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

# Appendix

## A    Distributions of A-TSI and M-TSI on ImageNet

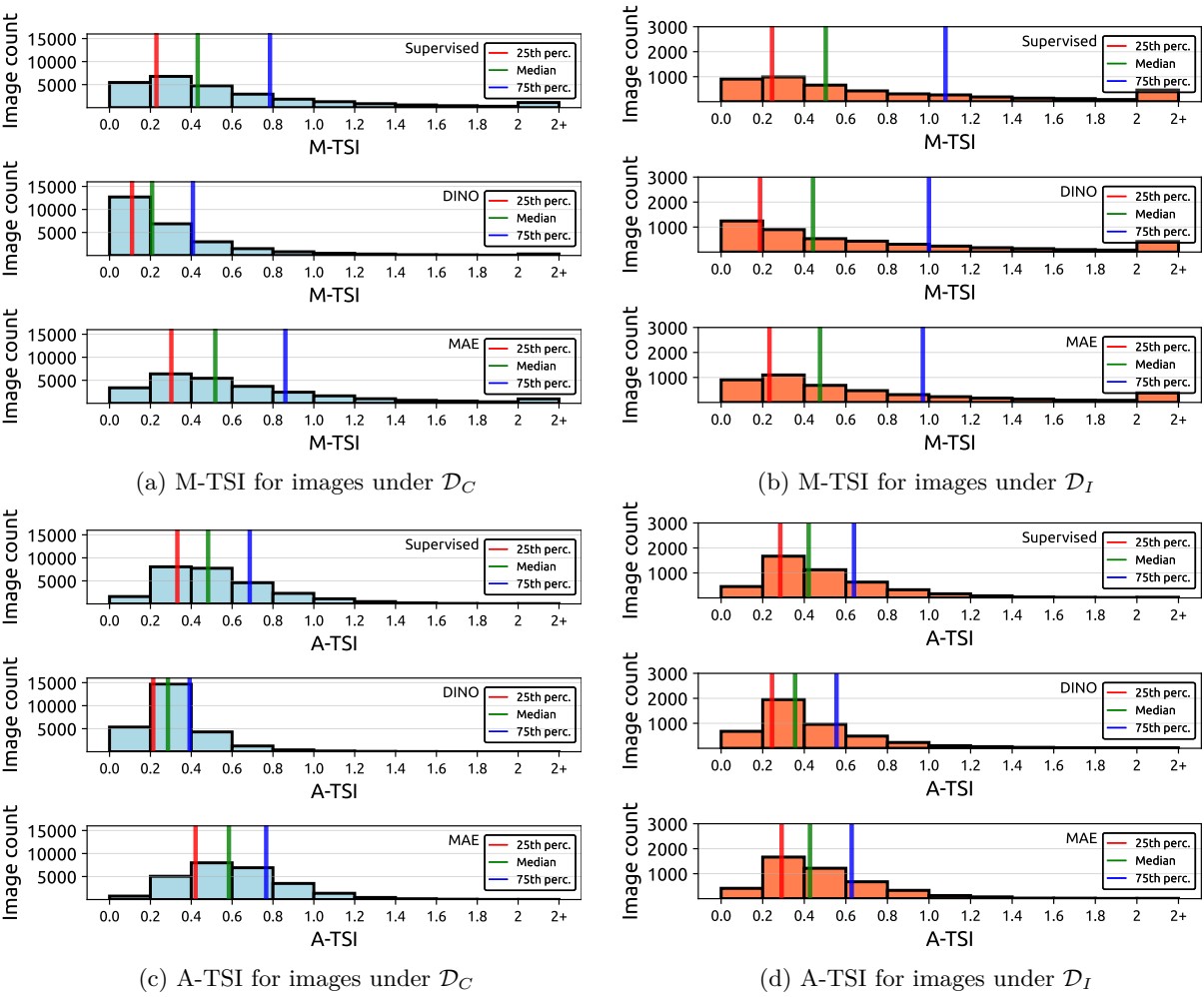

(a) M-TSI for images under $\mathcal{D}_C$

(b) M-TSI for images under $\mathcal{D}_I$

(c) A-TSI for images under $\mathcal{D}_C$

(d) A-TSI for images under $\mathcal{D}_I$

Figure 11: Histograms of M-TSI and A-TSI for ViTs trained using supervised learning, DINO, and MAE on $\mathcal{D}_C$ and $\mathcal{D}_I$. Images with a TSI value greater than 2 are grouped into the $[2, 2+]$ bin for clarity.

## B  On image masking, token discarding, and missingness bias

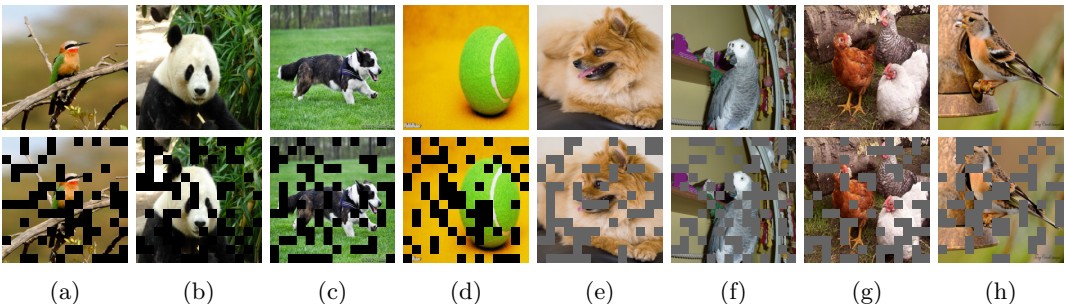

Figure 12: Example input images (top) and their masked versions (bottom). While both the original and masked images are correctly classified by ViTs when masked tokens are discarded, the masked images are misclassified by ResNet-50, a prominent CNN architecture, into plausible but incorrect categories due to missingness bias. The initial and masked predictions for the images are: (a) bee-eater → boa, (b) panda → soccer ball, (c) corgi → doormat, (d) tennis ball → mousetrap, (e) Pomeranian → African gray, (f) African gray → quill, (g) hen → spotlight, and (h) brambling → crossword puzzle.

Missingness bias in convolutional neural networks (CNNs) arises when features are removed or masked in a way that introduces unintended distortions in model predictions (Balasubramanian & Feizi, 2023; Jain et al., 2022). Since CNNs rely on convolutions that operate over a spatially contiguous image, they cannot naturally ignore missing regions (LeCun et al., 1998). Instead, missing pixels must be replaced with approximations, such as blacking them out, adding noise, or using a blurred region. These approximations inadvertently introduce biases because the model learns to associate the masking pattern itself with certain predictions, rather than relying on the remaining unmasked features. This effect not only affects model predictions but also interpretability techniques such as LIME, where missingness is used to infer feature importance. In contrast, ViTs offer a more natural implementation of missingness through token discarding. Since ViTs process images as a set of non-overlapping tokens rather than a spatially continuous grid, individual tokens corresponding to specific image regions can be discarded entirely without causing missingness bias. As a result, ViTs mitigate missingness bias and enable more reliable model debugging, making them particularly advantageous for feature attribution and interpretability studies.

In Figure 12, we provide several example cases where images have their content partially masked, demonstrating the impact of missingness bias on CNNs. While the original images are correctly classified by ResNet-50, their masked counterparts are misclassified into plausible but incorrect categories. This occurs because the missing regions alter key visual features, leading the model to rely on incomplete or misleading cues. The figure illustrates how different objects, such as animals and everyday items, shift in classification due to the absence of crucial details. These examples highlight the susceptibility of CNNs to missing information (i.e., masking) and the importance of understanding how they handle occlusions and data sparsity compared to token discarding in ViTs.

## C  Ablation studies

Table 4: Mean (standard deviation) prediction confidence changes for the correct class after masking a varying number of tokens according to GradCAM and Token Influence.

| Number of Tokens | Supervised | | DINO | | MAE | |
|---|---|---|---|---|---|---|
| | GradCAM | Token Influence | GradCAM | Token Influence | GradCAM | Token Influence |
| 1 | 0.02 (0.04) | 0.06 (0.08) | 0.02 (0.05) | 0.05 (0.08) | 0.01 (0.02) | 0.03 (0.05) |
| 3 | 0.04 (0.06) | 0.09 (0.12) | 0.04 (0.07) | 0.09 (0.13) | 0.02 (0.04) | 0.05 (0.08) |
| 5 | 0.06 (0.08) | 0.11 (0.14) | 0.05 (0.09) | 0.11 (0.16) | 0.03 (0.04) | 0.05 (0.09) |
| 10 | 0.08 (0.10) | 0.14 (0.16) | 0.06 (0.11) | 0.15 (0.20) | 0.04 (0.05) | 0.07 (0.10) |
| 20 | 0.11 (0.13) | 0.20 (0.19) | 0.09 (0.14) | 0.21 (0.24) | 0.04 (0.06) | 0.08 (0.12) |

### C.1  GradCAM to identify token influence

We compare our proposed method for quantifying token influence against GradCAM, a widely used technique for visualizing and interpreting neural network predictions. Specifically, we generate both GradCAM heatmaps and token influence maps for each correctly classified image ($\mathcal{D}_C$). Next, we systematically mask the tokens identified as most important according to each method. For GradCAM, we mask the tokens corresponding to the highest activation values in the GradCAM heatmaps (as illustrated in Figure 12). Similarly, for the token influence maps, we mask the tokens with the highest importance scores. We repeat this masking process for varying numbers of tokens, using $n \in \{1, 3, 5, 10, 20\}$. For each value of $n$, we record the resulting change in the model's prediction confidence for the correct class after the masking is applied. Finally, we compute the average change in prediction confidence across all images and report average prediction changes as a measure of how sensitive the model's output is to the tokens identified by each interpretability method in Table 4.

As can be seen in Table 4, masking tokens based on our proposed token influence method consistently leads to larger decreases in prediction confidence for the correct class compared to GradCAM, across all models and token counts.

For all three models—Supervised, DINO, and MAE—the drop in confidence grows as more tokens are masked, indicating that both methods correctly identify influential tokens. However, the proposed token influence approach results in noticeably higher prediction shifts, especially when masking a larger number of tokens. This trend highlights that our method more effectively captures tokens critical to the model's decision-making process compared to GradCAM, thereby demonstrating a more precise identification of influential tokens for predicting the correct class.

### C.2  GradCAM and attention maps as alternatives to token discarding

As discussed in Section 3.1.1, generating token influence maps involves discarding each token individually and measuring the resulting change in the model's prediction for the correct class. This process requires multiple forward passes and can become infeasible in resource-constrained settings. To mitigate this, we investigate whether attention maps can serve as a more efficient substitute, as they can be obtained in a single forward pass without repeated token masking.

We generate attention maps for all images under $\mathcal{D}_C$ and $\mathcal{D}_I$ and for all three models. Using these attention maps, we compute M-TSI and A-TSI scores in the same manner as with token influence maps. We then assess the degree to which TSI scores using attention maps correlate with TSI scores calculated using token influence maps by calculating Pearson's correlation coefficient. These results are presented in Table 5. Furthermore, we also use the previously generated GradCAM heatmaps to perform the same type of analysis, and present those results in Table 5 as well.

Our results reveal varying levels of correlation across models. For the supervised model, we observe a weak positive correlation, while the MAE model exhibits a moderate positive correlation. In contrast, DINO demonstrates a strong positive correlation between attention-based and token influence-based TSI

Table 5: For both M-TSI and A-TSI, we compute the correlation between TSI scores derived from different interpretation methods across all images in $\mathcal{D}_C$ and $\mathcal{D}_I$. The first table shows the correlation between TSI scores obtained from Token Influence Maps and those from attention maps. The second table shows the correlation between Token Influence Maps and GradCAM patch attributions.

| Subset | Metric | Attention Maps | | | GradCAM | | |
|--------|--------|------------|------|------|------------|------|------|
| | | **Supervised** | **DINO** | **MAE** | **Supervised** | **DINO** | **MAE** |
| $\mathcal{D}_C$ | M-TSI | 0.308 | 0.613 | 0.288 | 0.213 | 0.331 | 0.201 |
| | A-TSI | 0.228 | 0.806 | 0.329 | 0.084 | 0.023 | 0.007 |
| $\mathcal{D}_I$ | M-TSI | 0.235 | 0.591 | 0.395 | 0.101 | 0.285 | 0.266 |
| | A-TSI | 0.168 | 0.825 | 0.584 | -0.006 | 0.035 | -0.001 |

scores. These findings suggest that attention maps may serve as a suitable proxy for token influence maps in discriminative self-supervised models such as DINO, but are less reliable for supervised and generative self-supervised models such as MAE. Overall, attention maps offer efficiency gains but do not consistently approximate token influence across models, and should not be relied upon in isolation for fine-grained spurious correlation detection. Moreover, although we explored the use of attention maps as a more efficient alternative to token influence maps, our results show that this substitution is model-dependent and does not consistently approximate token influence across different architectures.

In contrast, using GradCAM heatmaps yields weak to negligible correlation with token influence-based TSI scores across all models. This indicates that GradCAM highlights different regions than those identified as influential by token masking. As a consequence, we discover that GradCAM is not suitable as a proxy for token influence in the context of spurious correlation analysis.

### C.3 Perturbation-based core vs. non-core feature evaluation

To further validate the proposed token-based spuriosity estimation, we perform an ablation inspired by the works of Singla & Feizi (2021) and Singla et al. (2022) on core and non-core feature analysis using perturbation-based evaluation. In particular, we adopt a variant of perturbation-based sensitivity analysis where semantic regions are explicitly corrupted to assess their influence on model predictions.

Instead of computing token importance via token discarding, we rely on pre-defined *core* and *non-core* masks and evaluate model reliance through controlled perturbations. Given an input image $\mathtt{X}$, we construct two perturbed versions:

- $\mathtt{X}^{(\text{core})}$: Gaussian noise is applied only to regions corresponding to the core mask.

- $\mathtt{X}^{(\text{non}-\text{core})}$: Gaussian noise is applied only to regions corresponding to the non-core mask.

In this scenario, the non-core mask refers to the areas in each image that are outside the core mask area. For the Gaussian noise, we follow the setup in Singla et al. (2022) and use $\sigma = 0.25$.

We then quantify the change in prediction confidence for the ground-truth class $c$, similar to the setup described in Section 3.1.1:

$$\Delta_{\text{core}} = g(\mathtt{X})_c - g(\mathtt{X}^{(\text{core})})_c, \quad \Delta_{\text{non-core}} = g(\mathtt{X})_c - g(\mathtt{X}^{(\text{non-core})})_c \tag{3}$$

Based on these measurements, we define a perturbation-based spuriosity score analogous to TSI, which we call noise spuriosity index (NSI):

$$\text{NSI} = \frac{\Delta_{\text{non-core}}}{\Delta_{\text{core}}} \tag{4}$$

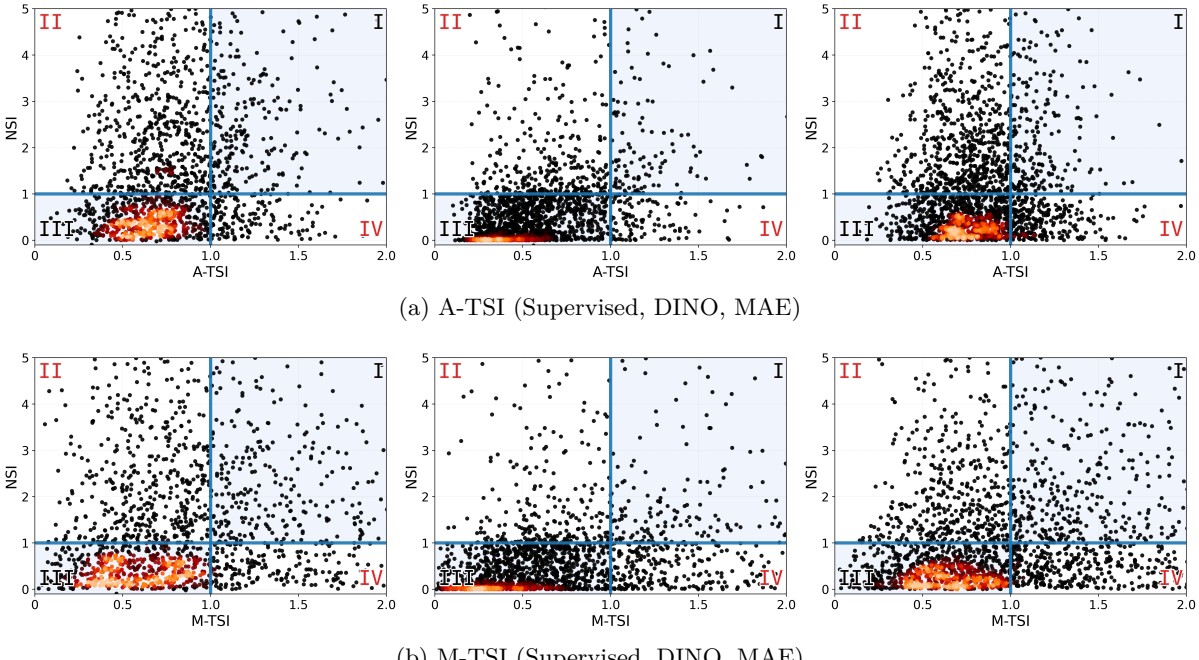

(a) A-TSI (Supervised, DINO, MAE)

(b) M-TSI (Supervised, DINO, MAE)

Figure 13: Quadrant-based comparison between the token-based and perturbation-based spuriosity indices across models (Supervised, DINO, MAE) and metrics (A-TSI, M-TSI). Each point represents an image, positioned according to the TSI and NSI estimates. Quadrants I and III correspond to agreement zones, where both methods consistently indicate either the presence or absence of spurious correlations. Quadrants II and IV correspond to disagreement zones, where the two methods provide conflicting assessments of feature reliance. Overlapping data points are highlighted using colormaps.

This formulation follows the same intuition as our token-based SI: if perturbations in non-core regions induce larger prediction changes than perturbations in core regions (NSI > 1), the model is likely relying on spurious features.

This ablation serves two purposes. First, it provides a direct comparison between our token-level attribution mechanism and a mask-based perturbation, where core and non-core masks are obtained via human-annotated saliency. Second, it allows us to evaluate whether the observed spuriosity trends are consistent across different feature importance estimation paradigms.

Since core and non-core masks are not available for all images in ImageNet, we restrict our analysis to a subset of annotated samples from Singla & Feizi (2021). We select 20 classes, resulting in a total of 4,375 images. For this subset, we compute TSI and NSI using both the proposed token-based method described in Section 3.1.1 and the perturbation-based variant described above, and present the comparison in Figure 13.

For clarity, we divide the figure into four quadrants:

- **Quadrant I** (TSI > 1, NSI > 1): Both methods indicate the presence of spurious correlations, suggesting an agreement that the model relies on non-core features.

- **Quadrant II** (TSI ≤ 1, NSI > 1): The perturbation-based method indicates spurious correlations, while the token-based method does not.

- **Quadrant III** (TSI ≤ 1, NSI ≤ 1): Both methods indicate reliance on core features, suggesting that predictions are primarily based on semantically relevant regions.

- **Quadrant IV** (TSI > 1, NSI ≤ 1): The token-based method indicates spurious correlations, while the perturbation-based method does not.

Table 6: Percentage of images falling into agreement (Quadrants I + III) and disagreement (Quadrants II + IV) between the token-based and perturbation-based SI across different models. Agreement corresponds to cases where both methods consistently indicate either the presence or absence of spurious correlations, while disagreement corresponds to cases where the two methods provide conflicting assessments of feature reliance.

| SI Methods | Quadrants | Supervised | DINO | MAE |
|---|---|---|---|---|
| A-TSI and NSI | Agreement (I + III) | 59.5% | 79.5% | 65.8% |
| | Disagreement (II + IV) | 40.5% | 20.5% | 34.2% |
| M-TSI and NSI | Agreement (I + III) | 62.9% | 77.3% | 67.0% |
| | Disagreement (II + IV) | 37.1% | 22.7% | 33.0% |

Based on Figure 13, the distribution of agreement and disagreement regions across quadrants is summarized in Table 6. Overall, we observe that the majority of samples lie in Quadrants I and III, corresponding to cases where the two methods agree. Across models and metrics, agreement ranges from approximately 59% to 79%, indicating a consistent level of alignment between the proposed TSI and the perturbation-based variant.

Notably, DINO once again shows the highest agreement (up to 79.5% for A-TSI and 77.3% for M-TSI). In contrast, the supervised and MAE models show moderately lower agreement, though still maintaining the majority of samples in agreement regions.

These results show that, despite relying on fundamentally different mechanisms, both approaches capture broadly consistent signals regarding the model's reliance on core versus non-core features, while also revealing non-negligible disagreement that reflects their complementary inductive biases. In particular, the perturbation-based method relies on mask quality and noise-induced perturbations which may introduce distributional artifacts, whereas the token-based method measures feature importance through structural removal within the model's native input space. Consequently, each method may emphasize different signals when assessing spurious signals. We believe that these discrepancies reveal an interesting direction for future work, where jointly analyzing and reconciling these complementary perspectives could lead to a more comprehensive understanding of model behavior and their reliance on spurious correlations.

### C.4  Attention maps as alternatives to manual annotation

To explore scenarios where annotations are not available or practical, we investigate whether attention maps can serve as an alternative for identifying objects of interest and quantifying spurious correlations, since attention maps have been shown to highlight objects of interest in prior research efforts, particularly for models trained in a self-supervised fashion. As such, instead of using annotations to select the object of interest and to identify spurious correlations based on tokens that lie inside the bounding box ($B_{in}$) and outside ($B_{out}$), we leverage attention maps generated by the model. We progressively select the top 5, 10, 20, 40, and 80 tokens from the attention maps and use them as $B_{in}$. By doing so, we compute M-TSI and A-TSI using the token influence maps without relying on bounding boxes, thereby alleviating the need for explicit annotations.

Table 7: For all images in $\mathcal{D}_C$ and $\mathcal{D}_I$, the correlation is calculated between TSI scores obtained using ground-truth bounding boxes and TSI scores obtained using high-attention regions as a substitute (for bounding boxes).

| Subset | Metric | Supervised | DINO | MAE |
|---|---|---|---|---|
| $\mathcal{D}_C$ | M-TSI | 0.487 | 0.200 | 0.429 |
| | A-TSI | 0.681 | 0.791 | 0.835 |
| $\mathcal{D}_I$ | M-TSI | 0.568 | 0.482 | 0.438 |
| | A-TSI | 0.703 | 0.668 | 0.731 |

We then examine the correlation between the best-case scenario of TSI scores calculated using attention-based $B_{in}$ selections and those computed using annotated bounding boxes. In Table 7, we report Pearson's correlation coefficients computed across all images under $\mathcal{D}_C$ and $\mathcal{D}_I$.

We observe that M-TSI, which is highly sensitive to the single-largest attention score, exhibits poor correlation with TSI scores computed using token influence maps. In contrast, A-TSI, which aggregates contributions across all tokens in Bin and Bout, demonstrates moderate positive correlation ($r > 0.6$) and strong positive correlation ($r > 0.7$). This finding indicates that A-TSI scores computed from attention maps as a method of token selection instead of manual annotation have the potential to be a proxy for identifying spurious correlations and assessing model focus, particularly in scenarios where annotations are unavailable. That said, relying solely on attention maps for token selection remains insufficient to fully capture token influence (especially for M-TSI), and care should be taken when interpreting such results, especially in models where attention distributions do not reliably reflect model reasoning.

### C.5 Limitations of using attention maps as alternatives in the TSI pipeline

In this ablation study, we employ GradCAM and attention maps as alternative proxies for different stages of the TSI computation pipeline. While such measures offer a degree of computational convenience, as they can be obtained without additional forward passes, the conclusions of these ablation studies should be interpreted in light of an important limitation regarding attention maps.

Although attention maps are frequently used to visualize token interactions in ViTs, several studies have shown that raw attention weights should not be interpreted as direct indicators of feature importance. In deep transformer architectures, information is progressively mixed across layers and propagated through residual connections, such that attention weights alone do not reliably reflect the contribution of individual tokens to the final prediction (Abnar & Zuidema, 2020). While our experiment of using attention maps as an alternative to token influence maps explores the potential of leveraging inherent ViT mechanisms to reduce computational overhead, we do not advocate for attention maps to be used as a direct replacement for token influence maps. We recognize the limitation that attention maps may not faithfully represent token-level importance across different model architectures and training settings.

We also acknowledge that the concept of using attention maps as a ground truth proxy should be approached with caution. In a recent study by Darcet et al. (2023), attention maps of several modern ViTs have shown to be dominated by outlier tokens and internal representation artifacts, ultimately obscuring semantically meaningful regions of an image. While this phenomenon was not witnessed in our experiments as we did not have the specific model variants for which such outlier dominance has been reported, this is to be noted that attention maps may not be the best-suited alternative for manual annotations when computing TSI scores.

For these reasons, we do not intend attention maps to be used as a direct substitute for token discarding–based influence estimation; rather, we suggest that attention maps are best viewed as qualitative visualizations rather than faithful, quantitative explanations of model behavior.

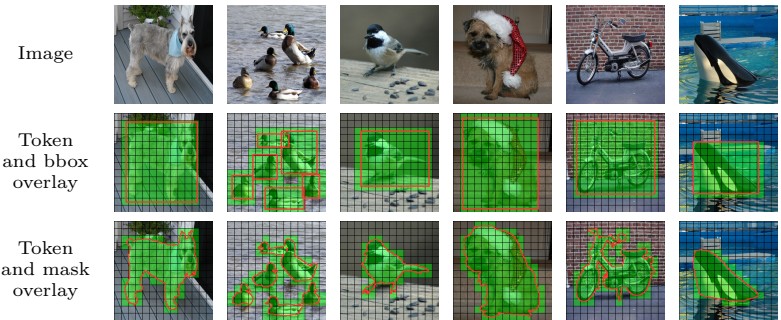

Figure 14: (Top) Example images, together with their corresponding (middle) bounding box annotations and (bottom) segmentation masks highlighting the object of interest. Annotation outlines are shown in red, and the ViT patches corresponding to the annotated regions are highlighted in green. Object regions defined by segmentation masks align closely with their corresponding bounding box annotations.

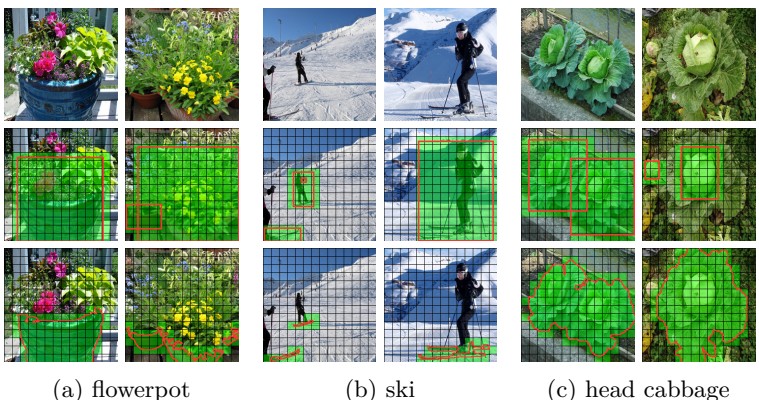

(a) flowerpot          (b) ski          (c) head cabbage

Figure 15: (Top) Examples from the ImageNet segmentation subset, grouped by class, together with their corresponding (middle) bounding box annotations and (bottom) segmentation masks highlighting the object of interest. Annotation outlines are shown in red, and the ViT patches corresponding to the annotated regions are highlighted in green. These bounding box annotations show inconsistent object coverage across images of the same class, while segmentation masks provide a more consistent semantic definition. For the same images, the object regions defined by segmentation masks differ substantially from bounding box annotations, being more tightly constrained than bounding box annotations.

## D    Using segmentation masks instead of bounding boxes

During dataset analysis, we observed that a portion of the segmentation annotations reflect a different interpretation of the target class compared to the ImageNet bounding box annotations, leading to inconsistencies between the two annotation sources (examples are shown in Figure 15). As discussed in Section 4, such discrepancies are also indicative of label inconsistency in the ImageNet dataset, where the definition of the object of interest may vary across images. In this context, semantic segmentation annotations provide a more fine-grained and explicit definition of the object region (see Figure 14), which can help identify and potentially mitigate ambiguities arising from loosely and inconsistently defined bounding boxes.

To assess how the choice of annotation affects the resulting TSI scores, we compute a pairwise TSI difference for each image, defined as $\text{TSI}_{\text{mask}} - \text{TSI}_{\text{bbox}}$. Across all models evaluated in our experiments, we observe that the use of segmentation masks consistently yields slightly higher M-TSI scores than bounding box annotations (see Figure 16). As shown in Figure 16, the pairwise difference between the two settings is predominantly positive. The distribution of these differences is concentrated in bins slightly above zero, indicating a modest increase in M-TSI when segmentation masks are used instead of bounding boxes. This behavior can be explained by the way M-TSI is computed. Recall that M-TSI is defined as the ratio of the

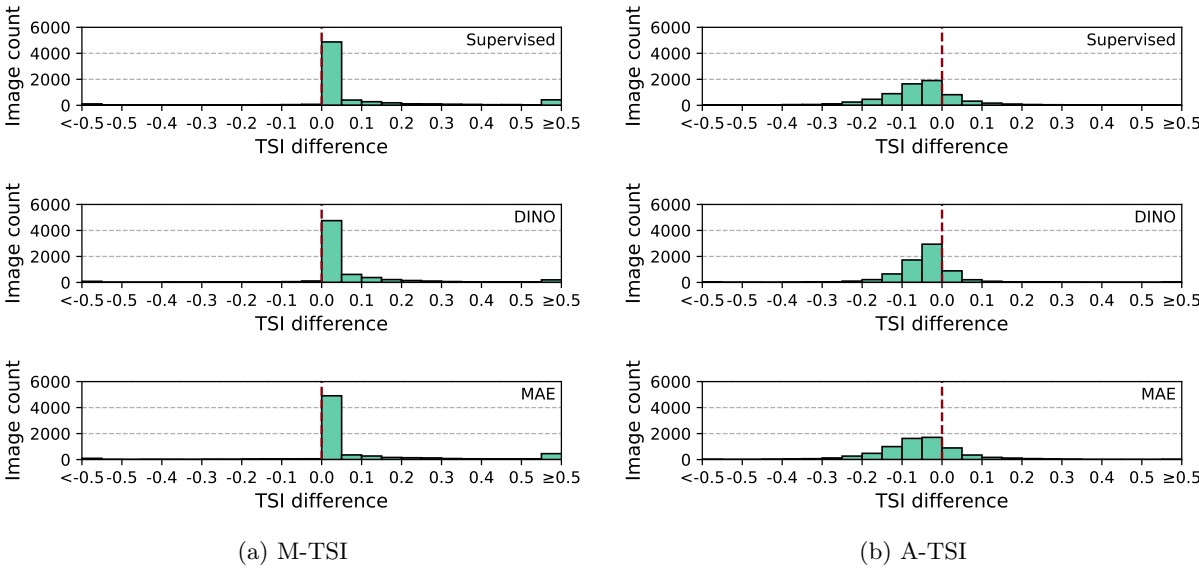

(a) M-TSI $\qquad$ (b) A-TSI

Figure 16: Histogram illustrating the pairwise differences between TSI scores computed using bounding boxes and segmentation masks across the three models (Supervised, DINO, and MAE). The difference is defined as $\text{TSI}_{\text{mask}} - \text{TSI}_{\text{bbox}}$ and is shown separately for (a) M-TSI and (b) A-TSI.

maximum token influence outside the annotated region, $\text{B}_{\text{out}}$, to the maximum token influence inside the region, $\text{B}_{\text{in}}$ (see Eq. 2). Compared to bounding boxes, segmentation masks typically define a more restrictive and precise set of foreground tokens. As a result, the set of tokens defining the object region originating from segmentation masks may exclude highly influential tokens that would otherwise be included under a looser bounding box annotation. This exclusion can reduce the maximum influence within $\text{B}_{\text{in}}$, thereby increasing the M-TSI value.

In contrast, we observe the opposite trend for A-TSI when computing the pairwise difference. When using segmentation masks, A-TSI scores tend to decrease compared to the bounding box–based setting as shown in Figure 16. Since A-TSI aggregates token influence using an averaging operation over all tokens in $\text{B}_{\text{in}}$ and $\text{B}_{\text{out}}$ (see Eq. 1), the reduced number of tokens defining $\text{B}_{\text{in}}$ under segmentation masks leads to a higher average influence within the annotated region when more of the background is omitted from the annotation.

Taken together, these results suggest that segmentation masks provide a more tightly constrained definition of the object's core region, which directly affects how different TSI variants respond to the annotation granularity. In particular, while M-TSI becomes more sensitive to the exclusion of highly influential patches near object boundaries, A-TSI appears to benefit from a more accurate definition of the object interior, making it more reliable when the core region is precisely defined. This highlights the importance of annotation choice when interpreting TSI scores and underscores the complementary behavior of M-TSI and A-TSI under different annotation regimes.

# E   Correlation of A-TSI with M-TSI

To explore the relationship between the TSI scores, we provide Figure 17, which illustrates the correlation between A-TSI and M-TSI across all three models on ImageNet, using both $D_C$ and $D_I$ separately. The corresponding $R^2$ values fall within the range of low to moderate positive correlations, which means that although there is some relationship between A-TSI and M-TSI, it is not particularly strong.

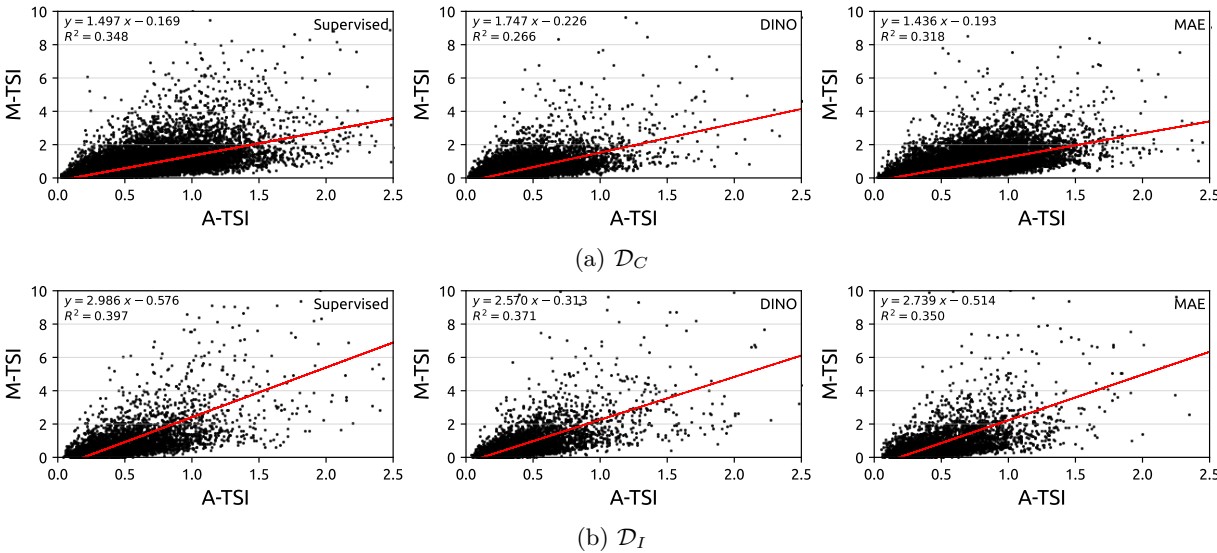

(a) $\mathcal{D}_C$

(b) $\mathcal{D}_I$

Figure 17: Scatterplots depicting the relationship between M-TSI and A-TSI for all images in (top) $D_C$ and (bottom) $D_I$. For each graph, the $R^2$ value is provided in the top left, and the model information is displayed in the top right.

