# OpenReview forum: "Token-Based Detection of Spurious Correlations in Vision Transformers"
_TMLR — Accepted by TMLR_

### Review · Reviewer_1W63 · 2025-12-22

**Summary Of Contributions:**

This work first introduces a simple method to detect image token importance by removing image tokens one by one to see the influence on classification accuracy. Then based on the importance score, the authors propose two metrics to evaluate the spurious correlations and provide an analysis on supervised and self-supervised models.

Strength:
- Well-formatted and easy to follow.
- Application to breast cancer detection is interesting.

Weakness:
- The method and metrics are quite simple, and lack theoretical analysis on why the method and metrics work, which makes the metrics less convincing.
- The findings are not surprising and lack deep analysis. For example, why does "DINO appear to be the more robust ViT", but for classification accuracy, MAE is actually higher than DINO?
- Only Imagenet is used in the experiments. More classification datasets should be used to validate the proposed metrics.
- Where does the bounding box come from? If it is generated from a pretrained model, is it reliable enough?

**Audience:**

Yes

**Audience Explanation:**

The topic is of interest to the TMLR audience.

**Claims And Evidence:**

Yes

**Claims Explanation:**

Method and experiment settings are mostly clearly described.

**Requested Changes:**

See above weakness.

---

> ### Author Response · Authors · 2026-01-09
>
> We are happy to hear that the manuscript was easy to follow and that the reviewer found the case study on breast cancer detection interesting. We would like to thank the reviewer for their careful reading of the manuscript and for the constructive feedback provided. In what follows, we will detail our proposals for adjustments according to the reviewer's suggestions.
>
> **Lack of theoretical justification:**
>
> We agree that the proposed token-based approach, including its use for detecting spurious correlations, is currently supported primarily by experimental evidence. We appreciate this observation and will use it as an opportunity to strengthen the theoretical grounding of the method by providing additional intuition, clarifying its underlying assumptions, and more explicitly discussing its limitations in the revised manuscript.
>
> **On the usage of ImageNet:**
>
> We chose ImageNet as it is the most widely used and well-understood large-scale image classification benchmark, making it particularly suitable for studying spurious correlations at scale. In contrast, smaller datasets such as CIFAR or MNIST often lack the complexity and diversity required for meaningful large-scale analysis, and observations drawn from such datasets may not generalize. Our evaluation spans several thousand ImageNet samples and is further complemented by a case study on the VinDr breast cancer detection dataset. We believe that this combination provides a meaningful and representative validation of the proposed metric, and we will clarify this motivation more explicitly in the revised manuscript.
>
> **Clarifying the Interpretation of results:**
>
> We will revise the manuscript accordingly and ensure that all statements are explicitly supported by and directly refer to the relevant tables and figures, so that the observations and conclusions are clearly grounded in the presented experimental evidence.
>
> **Bounding box annotations:**
>
> The bounding boxes used in our experiments are provided as part of the official ImageNet annotations. As such, they are human-annotated and widely used in prior work, making them a reliable source of object localization for large-scale analysis.
>
> **Proposed changes:**
>
> Given the feedback from the reviewer, we propose making the changes outlined below in the revised manuscript:
>
> - We will further ground the proposed method within a theoretical framework.
> - We will clarify in the manuscript the motivation for using ImageNet as the primary large-scale benchmark.
> - We will revise the manuscript to ensure that all statements are precise, well-supported by the presented experimental evidence, and clearly referenced to the relevant tables and figures where appropriate.
> - We will clarify the source of bounding box annotations.
>
> Apart from the changes requested by the reviewer, we will also make the following major additions:
>
> - Following the feedback of Reviewer 4Nto, we will conduct additional experiments using segmentation masks and provide direct comparisons with bounding box–based annotations. We will also revise the manuscript to explicitly acknowledge the limitations of using attention maps as a source of token importance and clarify their role in the ablation study.
> - Following the review of bqGC, we will revise and reorganize the abstract and introduction to better align them with the experimental findings, and expand the related work section with a dedicated discussion on token discarding and masking for interpretability.
>
> In summary, we believe we can address the reviewer's concerns through the revisions outlined above. We have responded to each comment with concrete proposed changes. If any of these revisions do not align with the reviewer's expectations, we would be happy to discuss further adjustments. Given the scope of the revisions, we plan to submit the revised manuscript within the week of 19 January and hope this timeline is acceptable.

---

### Review · Reviewer_4Nto · 2025-12-28

**Summary Of Contributions:**

This paper proposes two token-based metrics, M-TSI and A-TSI, as tools for detecting spurious correlations in Vision Transformers (ViTs). These metrics quantify a model's reliance on spurious signals by comparing the importance of tokens inside versus outside annotated bounding boxes. Through extensive experiments on ImageNet using both supervised and self-supervised ViTs, the authors provide insights into how different training methods influence such spuriosity.

**Audience:**

Yes

**Audience Explanation:**

The topic of spurious correlations is of significant interest to the machine learning community, especially as ViTs gain wider adoption. Analysis tools for diagnosing model behavior are valuable, and this paper offers some meaningful insights. Nevertheless, the major concerns outlined above should be addressed to strengthen the work.

**Broader Impact Concerns:**

There are no additional ethical concerns beyond those typically associated with AI transparency research.

**Claims And Evidence:**

No

**Claims Explanation:**

While the claims are generally supported by qualitative examples and analysis, several notable limitations weaken the evidence presented.
- The TSI metrics operate under the assumption that tokens inside bounding boxes represent “core features.” In practice, however, bounding boxes frequently contain substantial background content, particularly when they are large. This conflation of foreground and background tokens may distort spuriosity measurements. The use of segmentation annotations could be more appropriate for this problem and could also provide a more reliable baseline for evaluating other annotations such as bbox.
- The process for identifying influential tokens is computationally expensive. Moreover, the suggestion that attention maps could serve as a suitable alternative to token discarding is questionable, especially in light of findings from works such as “Vision Transformers need Registers,” which indicate that attention maps in larger ViTs can be significantly disrupted or dominated by outliers.
- Although the appendix explores using attention maps as a proxy for manual annotations, the authors do not adequately address well-known issues with attention maps—such as their vulnerability to outlier dominance in larger ViTs—which could compromise the reliability of the method. Also, the attention map is also influenced by the spurious correlation.

**Requested Changes:**

See the “Are the claims made in the submission supported by accurate, convincing and clear evidence?” section for details.

---

> ### Author Response · Authors · 2026-01-09
>
> We would like to express our gratitude to the reviewer for their thoughtful comments and efforts towards improving our manuscript, as well as their constructive feedback in highlighting its limitations. We believe that addressing these points will significantly strengthen the clarity, rigor, and positioning of the proposed work.
>
> **Bounding boxes and background content:**
>
> We agree with the reviewer that in certain cases (especially when objects of interest are diagonally oriented or loosely localized), bounding boxes may include substantial background content. We also agree that segmentation masks would be more appropriate for separating foreground from background tokens. However, segmentation annotations have historically been costly to obtain, and ImageNet officially provides bounding boxes, which enabled large-scale evaluation in our study. That said, as suggested by the reviewer, we believe the manuscript would benefit from a comparison between segmentation masks and bounding boxes. We will conduct additional experiments using segmentation annotations to assess their impact on TSI and report the results in the revised manuscript.
>
> **Computational cost:**
>
> Despite the seemingly high computational requirements, determining token importance for images with 196 tokens takes less than a second, even on decade-old consumer-grade GPUs such as TITAN X. Since token importance needs to be computed only once, the cost of obtaining it is negligible compared to model training and does not pose a practical bottleneck for large-scale evaluation. Moreover, our token importance method scales linearly with the number of tokens, in contrast to other methods, such as Shapley values, which consider an exponential number of token combinations.
>
> **On the usage of attention maps:**
>
> We would like to thank the reviewer for highlighting this important topic regarding the known weaknesses of methods that rely on attention maps. To clarify, in our work, neither the proposed method for calculating token influence nor the proposed TSI metric relies on attention maps. The method used for identifying important tokens directly measures changes in prediction confidence with token discarding, thereby assessing the model's reliance on individual tokens based on predictions.
>
> In the appendix, we provide ablations in which attention maps are used only as an alternative source of token importance for comparison purposes. We fully agree that attention maps can be affected by outlier dominance in larger ViTs and may themselves be influenced by spurious correlations. We will revise the manuscript to make this distinction clearer and to explicitly state that attention-based results are included for completeness rather than as a reliable substitute for the proposed token influence formulation.
>
> **Proposed changes:**
>
> Given the feedback of the reviewer, we propose making the following changes in the manuscript:
>
> - We will conduct additional experiments using segmentation masks and provide direct comparisons with bounding box–based annotations.
> - We will revise the manuscript to appropriately acknowledge the limitations of using attention maps as a source of token importance and clarify their role in our analysis.
>
> Apart from the changes requested by the reviewer, we will also make the following major additions:
>
> - Following the review of bqGC, we will revise and reorganize the abstract and introduction to better align them with the experimental findings, expand the related work section with a dedicated discussion on token discarding and masking for interpretability, and further strengthen the theoretical grounding of the proposed method.
> - Following the feedback of 1W63, we will clarify the motivation for using ImageNet as the primary large-scale benchmark, and explicitly state the source of the bounding box annotations in the revised manuscript and to ensure that all statements are precise, well-supported by the presented experimental evidence, and clearly referenced to the relevant tables and figures.
>
> With these revisions, we believe we can address the reviewer's comments thoroughly. We would be happy to discuss the appropriateness of revisions to ensure they align with the reviewer's expectations. Given the scope of the changes, we plan to submit the revised manuscript during the week of 19 January and hope that this timeline is suitable.

---

### Review · Reviewer_bqGC · 2026-01-04

**Summary Of Contributions:**

## Summary
This paper proposes a token-based approach to identifying non-core (spurious) features that influence a model’s decision-making. The central idea is to evaluate the token importances by masking/dropping tokens.

Based on this intuition, this work then introduces two empirical metrics: the Average Token Spuriosity Index (A-TSI) and the Maximum Token Spuriosity Index (M-TSI), and uses them for empirical studies.

## Strengths
1. The paper is generally well written and easy to follow.
2. Although token masking is widely used in efficiency and NLP, the contribution lies on the measurement framework of core/non-core features based on token discarding.

## Weaknesses

1. Comparisons to established baselines for identifying core vs. non-core features? The paper only analyzes their their own results on ImageNet but lack the comparisons with other methods, such as attribution-based methods etc.

2. Limited applicability to semantic features. The method relies on spatially localized tokens and bounding boxes, making it effective for localized visual artifacts but less suitable for semantic or non-local attributes (e.g., color or texture), and thus shares limitations with pixel- or patch-based approaches.

**Additional Comments:**

NA

**Audience:**

Yes

**Audience Explanation:**

NA

**Claims And Evidence:**

Yes

**Claims Explanation:**

NA

**Requested Changes:**

## Major.

Proper evaluation of token importance typically relies on a principled theoretical framework (e.g., Shapley value theory). Simply dropping, masking, or discarding individual tokens does not yield a correct attribution of importance, as the resulting credit assignment ignores interactions and cooperation among tokens. While the proposed approach may be empirically useful, it lacks a theoretical justification for why single-token ablation provides a faithful measure of token importance. This absence of theoretical grounding constitutes a major concern.


## Minor.

1. The findings presented in the abstract/introduction and later experiments seem not consistent? I would suggest to polish the abstract.

2. In the introduction, this paper states:
> Using the proposed approach, we also identify the problematic classes in the ImageNet dataset,...

This seems not consistent with the statement in the later experiments. I would suggest to polish/re-organise the findings.

3. typo. "or"--> "and"
> Region selection is typically guided by interpretability techniques such as GradCAM (Selvaraju et al., 2017), LIME (Ribeiro et al., 2016), or Integrated Gradients (Sundararajan et al., 2017).

4. In the related work. It would be nice if adding a brief paragraph regarding token discarding/masking method for interpretability.

5. In Fig. 3 & 4. what are $D_c$ and $D_L$ $D_I$? Captions can be improved for readability in a more self-contained sense.

6. Sec. 3.3.1. what is $g_theta (x)$?

7. lack of baseline comparisons for spurious correlation detection.

---

> ### Author Response · Authors · 2026-01-09
>
> We are grateful to the reviewer for their careful reading of the manuscript and for the thoughtful and constructive feedback, including the identification of its limitations. The comments were particularly helpful in clarifying the theoretical aspects of the work and in improving the overall presentation. Below, we address each comment in detail and outline the revisions we plan to make to strengthen the manuscript.
>
> **Lack of theoretical justification:**
>
> The reviewer is indeed correct that the proposed token-based method, as well as its usage for detecting spurious correlations, is demonstrated primarily through experimental evidence. We appreciate the reviewer's guidance on this point and further note that the proposed approach may share some conceptual similarities with certain principles underlying Shapley values. We will use this opportunity to more clearly articulate the theoretical grounding, provide additional theoretical intuition, and explicitly discuss the assumptions and limitations of the method in the revised manuscript.
>
> **Comparison to other methods for spurious correlations:**
>
> The proposed metric (TSI) can be employed using any method that produces token-based attributions, such as Grad-CAM, attention-based scores, or the method of Chefer et al., instead of the token influence formulation used in the main manuscript. In the appendix, we provide an ablation using Grad-CAM to demonstrate this flexibility.
> To the best of our knowledge, there is no straightforward method that directly yields a quantitative measure distinguishing core from spurious features in a comparable manner, unlike TSI, where TSI > 1 indicates spurious reliance (non-core contribution exceeding core contribution) and TSI < 1 indicates non-spurious reliance (core contribution exceeding non-core contribution), with the magnitude reflecting the degree of spuriousness or core dominance. Most existing attribution-based approaches are primarily qualitative or instance-level, and do not define a numerical criterion for spuriousness that could be directly compared against TSI.
> That said, if the reviewer has a specific attribution method or evaluation protocol in mind that they believe constitutes a suitable quantitative baseline, we would be happy to incorporate additional experiments and report comparative results in the revised manuscript.
>
> **Suitability for semantic or non-local attributes:**
>
> We agree with the reviewer that the proposed method is less suited for purely semantic or non-local attributes such as global color or texture cues, as it operates on spatially localized tokens. This limitation is shared by most pixel-, patch-, and token-based interpretability approaches. More broadly, identifying non-local or globally distributed features remains a challenging and largely open problem in the literature.
>
> **Proposed changes:**
>
> Given the feedback from the reviewer, we propose making the following adjustments in the manuscript:
>
> - We will further ground the proposed method within the framework of Shapley value theory.
> - We will revise and reorganize the abstract and introduction to better align them with the experimental findings.
> - We will expand the related work section with a dedicated discussion on token discarding and masking for interpretability, as suggested by the reviewer.
> - We will fix the remaining minor issues raised by the reviewer.
>
> Apart from the changes requested by the reviewer, we will also make the following major additions:
>
> - Following the feedback of Reviewer 4Nto, we will conduct additional experiments using segmentation masks and provide direct comparisons with bounding box–based annotations. We will also revise the manuscript to explicitly acknowledge the limitations of using attention maps as a source of token importance and clarify their role in the ablation study.
> - Following the feedback of Reviewer 1W63, we will clarify the motivation for using ImageNet as the primary large-scale benchmark, and explicitly state the source of the bounding box annotations in the revised manuscript and to ensure that all statements are precise, well-supported by the presented experimental evidence, and clearly referenced to the relevant tables and figures.
>
> We hope that the revision plans outlined above address the reviewer's concerns. We would welcome the opportunity to discuss any of these revisions to ensure they align with the reviewer's expectations. Given the scope of the changes, we plan to submit the revised manuscript during the week of 19 January and hope this timeline is acceptable.

---

### Author Response · Authors · 2026-01-25
**Summary of Revisions**

We would like to thank the reviewers once again for their feedback and guidance. We have made the proposed changes discussed in our previous communication and highlighted all changes in the manuscript using a teal color.

In what follows, we detail the changes that have been made in the paper.

### **Major Changes**

**Expanding the Related Work**

We have expanded the related work section to cover several additional areas, with clear identification of their relevance to our work. The related work section now highlights:

- Training-based methods to avoid spurious correlations
- Attribution methods to detect spurious correlations
- Datasets for investigating spurious correlations
- Occlusion-based methods to detect spurious regions
- Token discarding in ViTs

**Theoretical grounding of our method**

We made several important changes to improve the theoretical grounding of our method. These changes include:

- A clear definition of the notation and the proposed token influence method
- Clarification of how our method is positioned relative to existing approaches, highlighting that it falls under a leave-one-out attribution strategy
- A description of how our method relates to occlusion-based attribution
- Following the work of Lundberg & Lee (2017), Sundararajan & Najmi (2020), and Covert et al. (2021), we describe several relevant properties of the method
- A discussion of the importance of missingness bias and how our method avoids it

We have also changed the order of subsections in the Methodology section to enable a smoother reading flow.

**Using segmentation masks instead of bounding box annotations**

We conducted additional experiments using segmentation masks instead of bounding boxes to calculate TSI scores. We observe that, in most cases, the difference between segmentation- and bounding box–based TSI scores is less than 0.05, with M-TSI scores generally increasing and A-TSI scores decreasing due to tighter boundaries. However, we find that noticeable differences arise in some instances, typically when bounding boxes cover substantially larger regions than the corresponding segmentation masks (e.g., for diagonally oriented or sparsely shaped objects), leading to larger TSI deviations. Summary results for these additional experiments can be found in Section 4, and we provide further details and additional results in Appendix D.

**Further details on the usage of attention maps**

We clarified that attention maps and Grad-CAM are used only as ablation proxies and are presented as qualitative or exploratory comparisons. We do not imply that using them as ground-truth explanations is a feasible alternative. As pointed out by the reviewer, we also state the limitations of attention maps, citing prior work on their unreliability and distributional biases and artifacts.


### **Minor Changes**

- Made several changes across the main text and the appendix to improve readability
- Made necessary clarifications regarding the dataset used
- Improved the readability of unclear figure and table captions (Fig. 1, Table 1)
- Fixed several typos and instances of inconsistent wording throughout the manuscript

We appreciate the reviewers’ time and thoughtful feedback, welcome any additional comments or remarks, and hope they will find the revised manuscript satisfactory.

### References

Scott M Lundberg and Su-In Lee. A unified approach to interpreting model predictions. *Advances in neural information processing systems*, 30, 2017.

Mukund Sundararajan and Amir Najmi. The many shapley values for model explanation. In *International conference on machine learning*, pp. 9269–9278. PMLR, 2020.

Ian Covert, Scott Lundberg, and Su-In Lee. Explaining by removing: A unified framework for model explanation. *Journal of Machine Learning Research*, 22(209):1–90, 2021.

---

### Author Response · Authors · 2026-04-24
**Summary of camera ready changes-1**

Dear Action Editor, Dear Reviewers,

We would like to thank you once again for your valuable feedback throughout the review process. We have carefully revised the manuscript to address the remaining concerns. Below, we summarize the final changes made in response to the latest feedback:

- We have revised the abstract and introduction to more clearly position the contributions of the paper. In particular, we avoid overstating novelty and instead describe our work as a token-based diagnostic pipeline that combines leave-one-out token removal with simple yet effective metrics.

- (Section 3.1.1) We have condensed the discussion of Properties 1-5 into a single paragraph and clarified that these properties are inherited from the leave-one-out (LOO) framework, with appropriate citations to prior work (e.g., Lundberg & Lee, 2017; Covert et al., 2021).

- (Section 3.1.1) We acknowledge the contributions of Jain et al. (2022), particularly regarding missingness bias and the use of token removal as a principled way to represent feature absence, and clarify how our work builds upon these insights.

- (Section 3.1.2) We further emphasize the limitations of using bounding boxes as proxies for core features. We note that TSI measures reliance on features outside annotated regions rather than on ground-truth spurious features, and we also discuss alternative annotation sources such as segmentation masks and refer to experiments using segmentation masks in the appendix.

- (Section 4) We have revised aggregate claims (e.g., comparisons across training paradigms such as DINO vs. MAE) to use more precise language, making it clear that TSI reflects relative reliance on features outside annotated regions.

- (Section 4) We have added a short discussion addressing the observation that DINO exhibits lower TSI but does not always achieve higher accuracy than MAE. We clarify that lower reliance on non-core features does not necessarily translate directly to higher classification performance, as robustness to spurious features and overall discriminative power are not perfectly aligned.

- (Section 4) We provide a detailed class-level report for all ImageNet classes as supplementary material, highlighting those that consistently exhibit high M-TSI scores. While the main text presents representative examples to visually illustrate the identified failure modes and provide an intuitive understanding, this large-scale report offers a more systematic view of the prevalence of such cases across the dataset. Additionally, our findings can be investigated alongside existing class-level analyses of spurious correlations, such as those provided by Singla & Feizi (2021), which serve as a useful reference for understanding dataset-level biases. We have modified the text and highlighted these changes in the manuscript.

- (Section 7) We acknowledge the lack of ground-truth spurious correlation annotations and clarify that TSI serves as a proxy measure, and we discuss this as a limitation.

---

> ### Author Response · Authors · 2026-04-24
> **Summary of camera ready changes-2**
>
> - (New Ablation, Section C.3 in the Appendix) We agree that having a baseline comparison with TSI is important; however, prior work has primarily focused on dataset-level bias mitigation or training objectives, and, to the best of our knowledge, does not address per-sample scoring of spurious reliance in trained models at inference time, particularly in large-scale natural image settings such as ImageNet.
>
>   With that said, we believe the closest works to ours are those of Singla & Feizi (2021) and Singla et al. (2022), where the authors create core and non-core masks for a subset of the ImageNet dataset using robust models and human annotations, and propose Core Risk Minimization, an approach that relies on perturbing non-core regions to minimize spurious correlations. However, this approach is once again designed for training robust models and mitigating spurious correlations at the dataset or training level, not for quantifying whether a model relies on spurious correlations for a given input at inference time.
>
>   To address this, we adopt their formulation to construct a comparable metric and evaluation protocol, and to create a setup analogous to our TSI scoring system using a perturbation-based variant with core and non-core masks. In doing so, we compare the extent to which their perturbation-based approach and our token-discarding approach agree or disagree with each other.
>
>   Across models and metrics (M/A-TSI), we observe consistent agreement between the two approaches, with approximately 59% to 79% (depending on the model and M/A-TSI) of samples falling into agreement regions where both methods indicate either (i) the presence or (ii) absence of spurious correlations. The remaining cases show disagreement, which we attribute to differences in how feature importance is estimated (token-based vs. perturbation-based), as well as the sensitivity of perturbation-based methods to mask quality and noise-induced artifacts. We believe that these discrepancies reveal an interesting direction for future work, where jointly analyzing and reconciling these complementary perspectives could lead to a more comprehensive understanding of model behavior.
>
> - (General) We fixed several typos and improved the wording of multiple sentences throughout the manuscript to enhance clarity and readability.
>
>
> ### References
>
> Sahil Singla and Soheil Feizi. Salient ImageNet: How to Discover Spurious Features in Deep Learning?, 2021.
>
> Sahil Singla, Mazda Moayeri, and Soheil Feizi. Core Risk Minimization Using Salient ImageNet, 2022.

---

### Decision · Action_Editor_4osj · 2026-03-15

**Recommendation:** Accept with minor revision

**Additional Comments:**

Please make sure to include the changes I described above to make the contributions clear. Also, please address the requests of the reviewers that are left unaddressed:
- Baseline comparison: The GradCAM ablation compares token importance methods, not spurious correlation detection methods. It would be helpful to add comparison to at least one existing approach.
- DINO vs. MAE discussion. Reviewer 1W63 asked why DINO has lower TSI but lower accuracy than MAE. The revision doesn't address this. It would benefit to add a discussion paragraph.
- Prevalence of confounding cases. The authors identify four failure modes of the bounding box assumption and mention manually inspecting "several hundred images." A more quantitative analysis would be helpful.

**Audience:**

Yes

**Audience Explanation:**

The empirical findings in the paper on how different design decisions affect the level of spurious feature reliance is definitely a topic of broader interest to the ML community, and I believe the community will appreciate the findings. Additionally, the applications to medical imaging could also be of interest given the importance of this issue in applications of ML to medicine.

**Claims And Evidence:**

Yes

**Claims Explanation:**

The empirical work is good and the paper has large-scale experiments across three training paradigms, useful ablations, and a medical imaging case study. However, the authors over claim the novelty of their method. The abstract describes "a novel token-based method" and the methodology section presents formal properties as though they constitute theoretical contributions. However, the method is a leave-one-out attribution applied via token removal, which the authors themselves acknowledge falls under established LOO frameworks (Lundberg & Lee, 2017, Covert et al., 2021). The listed properties hold trivially for any LOO approach by construction. Moreover, the use of token discarding in ViTs to implement true feature absence was demonstrated by Jain et al. (2022), which the authors cite. The metrics are also quite straightforward ratios. This doesn't diminish the paper's empirical value, but definitely falls under over claiming, making the claims not accurate. Here are some suggestions to address this:

- Revise the abstract and introduction to position the novelty clearly. For example: "We present a token-based diagnostic pipeline for detecting spurious correlations in vision transformers, combining leave-one-out token removal with two complementary metrics..."

- Condense Properties 1–5 into a single paragraph noting these are inherited LOO properties, citing Covert et al. (2021), rather than presenting them as contributions.

- Acknowledge more explicitly emphasize Jain et al.'s missingness insight and the fact that their work is an application.

Two additional concerns: (a) TSI assumes bounding box contains the core features, but the authors identify multiple cases where this breaks down (Figure 8). Aggregate claims like "DINO exhibits fewer spurious correlations" should be softened to reflect that TSI measures reliance on features outside the annotated region, which is a proxy for spurious features. (b) There is no external validation of TSI against ground-truth spurious correlations. The authors should acknowledge this limitation.

Given that these are primarily reframing changes rather than new experiments/theory, I think the authors can do this with minor revisions, and it doesn't take away from the positives of the paper.

---

> ### Author Response · Authors · 2026-03-30
> **Author response to Action Editor and Reviewers**
>
> Dear Action Editor, Dear Reviewers,
>
> We sincerely thank you for the positive reception of our manuscript and for your valuable guidance. Your comments have helped us improve not only the overall quality of the paper, but also the clarity of the used terminology, the methodological description, and the overall presentation.
>
> We will carefully address all points raised by the Action Editor and submit a revised version of the manuscript within the allocated time frame.
>
> Once again, we greatly appreciate your time, effort, and constructive feedback throughout.
>
> Kind regards,
>
> Authors